# Study on Joint Distribution Mode and Evolutionary Game of Express Enterprises in Rural Areas

Hongxiang Zhao  and Meiyan Li *

College of Energy and Mining Engineering, Shandong University of Science and Technology, Qingdao 266590, China
* Correspondence: skd994322@sdust.edu.cn

**Abstract:** Express delivery in rural areas of China has many problems, such as high delivery cost and low efficiency. As an effective way to solve the difficulties of rural delivery, it is important to study the innovation and application of a joint distribution model. In the background, this paper takes express delivery enterprises in rural areas as the research object. First, it proposes to construct a three-level "county-town-village" joint distribution system in which e-commerce platforms participate. Next, it establishes an evolutionary game model of express delivery enterprise joint distribution alliance and solves it. Finally, the model is analyzed through numerical simulation. The results show that the distribution system of express delivery enterprises in rural areas is affected by excess returns, early input costs, operating costs, cooperation risks, penalty costs, learning and absorption capacity of enterprises and other factors. After introducing the rewards and punishments of e-commerce platforms as an independent influencing factor in the evolutionary game model, the shorter time for express companies to finally make cooperation strategies indicates that the rewards and punishments of e-commerce platforms have a positive significance in promoting the rapid development and stable operation of a rural logistics joint distribution system.

**Keywords:** rural express; joint distribution system; evolutionary game; system stability; numerical analysis

## 1. Introduction

### 1.1. The Current Situation of Rural Express Delivery

In recent years, the Internet coverage in rural China has increased. According to the 50th Statistical Report on the Development of the Internet in China released by CNNIC in Beijing, the number of Internet users in rural China reached 293 million by June 2022, with a penetration rate as high as 58.8%, which shows that more and more farmers have joined the ranks of the Internet [1]. At the same time, more express deliveries are being delivered to thousands of households in rural areas via e-commerce platforms. In 2021, the annual express delivery volume in China's rural areas is 37 billion, with a year-on-year growth of 23% [2].

Due to the influences of culture, geographical environment and regional development level in rural areas, the rural express delivery chain is extended or even disjointed. The traditional rural express logistics distribution mode has some problems, which has been a bottleneck restricting the development of rural logistics and the rural economy. The main reasons for the bottleneck are summarized as follows: First, the low level and single structure of rural basic transportation construction leads to the expansion of the scope of express delivery and the smaller distribution quantity per unit area. In 2021, the domestic rural express business volume only accounted for 1/5 of the total business volume, while the delivery route mileage accounted for 63% [3]. Second, the delivery capacity of the express terminal is insufficient, and the number of rural terminal delivery stations is also insufficient. Most of the station distribution stays at the township level, which makes it difficult to achieve full coverage of administrative villages, and the "last mile" distribution cannot be realized. Third, the express delivery enterprises themselves, and the government's supervision of rural express delivery enterprises, are insufficient and lack unified

management, resulting in poor delivery service quality. In general, the postal is complete but not fast, SF Express is fast but not complete, Jingdong is good but not big and Cainiao is big but not strong [4].

The development of joint distribution in rural areas is mainly reflected in two aspects [5]. For example, by improving the information service capability of the rural delivery logistics system, the communication between remote areas and the outside world has been opened [6]. In contrast, farmers transport agricultural products to urban markets in a timely and accurate manner [7]. In summary, the rural joint distribution system will further smooth the two-way circulation channels for consumer goods to go to the countryside and villages and for agricultural products to go to the city, which will help solve the problem of insufficient consumption facilities and poor sales channels in rural areas and facilitate the production and life of rural people. At the same time, the joint distribution system closely combines the development of the industry with the needs of the rural people for a better life [8].

As a distribution mode that can effectively reduce the distribution cost and improve the service level of distribution, joint distribution can effectively solve many problems in the current rural express distribution. Joint distribution requires cost sharing, income sharing, and highly close cooperation among cooperative subjects. However, at present, China's logistics industry is in the growth stage of rapid development, and it is difficult to coordinate among express delivery enterprises. Therefore, building a joint distribution system suitable for rural express delivery and exploring the decision-making behavior characteristics and stable state of all parties in the system are two urgent problems to be solved.

### 1.2. China's Rural Express Delivery Mode

According to China's rural express delivery undertaker, the modes are divided into self-distribution delivery mode, third-party delivery mode and joint distribution mode. The three modes of express delivery in rural areas have their characteristics in the application process.

The rural express delivery mode is mainly operated by e-commerce, which means that the e-commerce platform independently builds infrastructure, improves the distribution network and strengthens the independent supervision of warehousing, transportation and distribution by relying on its own platform advantages, and is mainly represented by Jingdong. Jingdong improves its "O2O" distribution network by virtue of its e-commerce platform and logistics network advantages and has formed a distribution pattern at the county and township levels [9].

The rural express delivery mode based on the third party means that both parties involved in the transaction entrust agents to third-party enterprises to complete the delivery tasks, mainly represented by Cainiao. Cainiao builds China's largest intelligent logistics platform by integrating several express delivery enterprises, such as Sandong and Yida, and makes use of the open and sharing advantages of resources, information and technology. This builds a rural distribution system of "county center +N village stations" [10].

Joint distribution mode refers to a consortium in which multiple enterprises cooperate to complete distribution tasks by integrating resources and optimizing the distribution process. In the process of rural express delivery, the hardware and software conditions such as information resources, transportation resources and storage resources are coordinated between e-commerce enterprises and express delivery enterprises to achieve the delivery purpose, which mainly represents the postal service. The postal service has built a rural distribution network covering towns and villages all over the country, which can break the barriers of rural logistics and solve the "last mile" distribution problem [11,12].

Many scholars have summarized the existing urban and rural logistics and distribution models and have put forward the general direction of distribution development according to the advantages and disadvantages of different models. Tao Chu made a comparative study of the main characteristics of three typical rural express delivery modes and analyzed their advantages, disadvantages and applications. It is proposed to improve the distribution mode system in rural areas and improve the service level oriented by customer service [13].

Wang Guihua and Raza Syed analyzed the joint distribution problem from the perspective of the commodity supply chain [14,15]. Yandong He conducts a study on urban logistics distribution patterns and uses a fuzzy analytic hierarchical process and fuzzy technology to perform research, which is effective and robust for JDA's joint distribution partner selection [16]. Pang H introduced several e-commerce logistics distribution modes and described the operating principle and application category of each e-commerce logistics distribution mode [17]. Xue Sun analyzes the current e-commerce logistics distribution model and its characteristics, puts forward how to choose the e-commerce distribution model based on qualitative analysis and quantitative analysis, and verifies the feasibility of the method through calculation examples [18]. Yao Keqin has three typical modes of rural e-commerce terminal logistics joint distribution in China and puts forward the decision-making path for the development of rural e-commerce terminal logistics joint distribution [19].

In view of the current situation of China's rural development, many scholars will adopt a hierarchical setting method to solve the problem of rural logistics and distribution, and the results have been fruitful. Zhou Yao proposed an urban–rural regional distribution model based on the joint distribution model. By integrating some urban and rural areas and treating them as a whole, a joint distribution center is established to replace the traditional county-level distribution center [20]. Zhang Xicai established a three-level logistics network system in counties and villages to get through the "last mile" to put rural logistics and agricultural logistics in an equally important position, and put forward the concept of rural logistics [21]. Chen Ying constructed the three-level logistics system of "county-town-village" in Xuzhou based on joint distribution and implemented countermeasures for the three-level logistics system [22].

Evolutionary game theory has been studied by many scholars due to its own characteristics of limited rationality. David K introduced the basic concepts and knowledge of evolutionary game theory and noted the connection between evolutionary game theory and traditional classical game theory based on the conceptual proposal [23]. Barari further discussed the development of evolutionary game theory [24]. With the deepening of related theoretical research, evolutionary game theory has been applied in an increasing number of fields. Na Zhang and Renbin Han proposed a three-way evolutionary game model of "government-member enterprise A-member enterprise B" under the guidance of government supervision [25,26]. Wan Xiaoyu and Luo Hanqi constructed a two-party evolutionary game model with the headquarters of express delivery enterprises as the main body under different subsidy policies [27,28]. Yu Xiaohui took rural express delivery as the research object and established a common distribution service mechanism at the end of rural express logistics under uncertain risks [29].

In addition, the results of evolutionary game research are summarized. Zhang Cheng [30], Liang Wen [31] and Xian Chuanzhi [32] discussed the stability of the co-evolution game between rural e-commerce and rural logistics under the government's poverty reduction strategy, and obtained the equilibrium strategy and stability factors through the analysis of the income matrix. The results show that the government's strategic choice of enterprises is an important factor affecting enterprise decision making, and the government's active promotion of strategy is conducive to the collaborative evolution of rural e-commerce and rural logistics. Li Changbing analyzed the strategy selection behavior of the government and logistics enterprises in the construction of rural logistics systems. The results show that the government can accelerate the development of the rural logistics market and actively guide the farmers' demand for logistics to achieve a mutually beneficial situation for the government and enterprises in the construction of the rural logistics market [33].

An increasing number of studies have been published on the combination of joint distribution and evolutionary game models, and the research conclusions of many scholars based on actual case data have certain value and reference. Based on the data of many postal express companies in rural Fuzhou, Zhang Qian explored the joint distribution mechanism and benefit distribution under the evolutionary game and put forward reasonable suggestions for the development of express delivery in rural areas [34]. Xu Meizhen simulated

the dynamic evolution process of the main distribution system under the self-organization environment and, with the participation of the government through the data of two express delivery companies in Fuzhou, demonstrated how the government's intervention can promote urban cold chain logistics enterprises to actively carry out joint distribution cooperation [35]. Based on the data of three cold chain enterprises in Beijing, Xu Zongping assigned the parameters in the evolutionary game model, showed the dynamic evolution process, and proposed the prerequisites suitable for joint distribution [36]. Based on the actual situation of five express delivery companies in Zhengzhou, and based on the conclusion of the evolutionary game model, Liu Shu constructed an incentive mechanism for joint distribution between express delivery enterprises [37].

In summary, scholars at home and abroad have achieved fruitful theoretical results in the analysis of the joint distribution mode and the evolutionary game of the participants, but there are few studies that combine the evolutionary game analysis in the joint distribution with rural express delivery. This paper takes express delivery in rural areas of China as the research object with the purpose of building a reasonable mode of joint distribution in rural areas and maintaining the stability of a joint distribution alliance. First, combined with the current situation of express delivery in rural China, the three-level joint distribution mode of "county-town-village" with the participation of e-commerce platforms was introduced. Next, through the evolutionary game model to study the decision-making behavior characteristics of the parties involved in joint distribution and its stable state, we put forward a stable strategy for the development of a joint distribution alliance. Finally, the numerical analysis of the example is carried out to further verify the scientific and rational results of the research.

## 2. Model Construction and Solution

### 2.1. Construction of a Three-Level Joint Distribution System of Rural Express "County, Township and Village"

By exploring the status quo of express delivery in China's rural areas and combining the study of the joint distribution mode, a three-level "county-town-village" joint distribution system of rural express involving e-commerce platforms is constructed, as shown in Figure 1. The construction of the three-level joint distribution system of "county-town-village" mainly includes three parts: county-level distribution centers in common, township-level transit stations and village-level service points. The county-level distribution centers in common are very important, and they shoulder the functions of scheduling, warehousing, transportation and information collection.

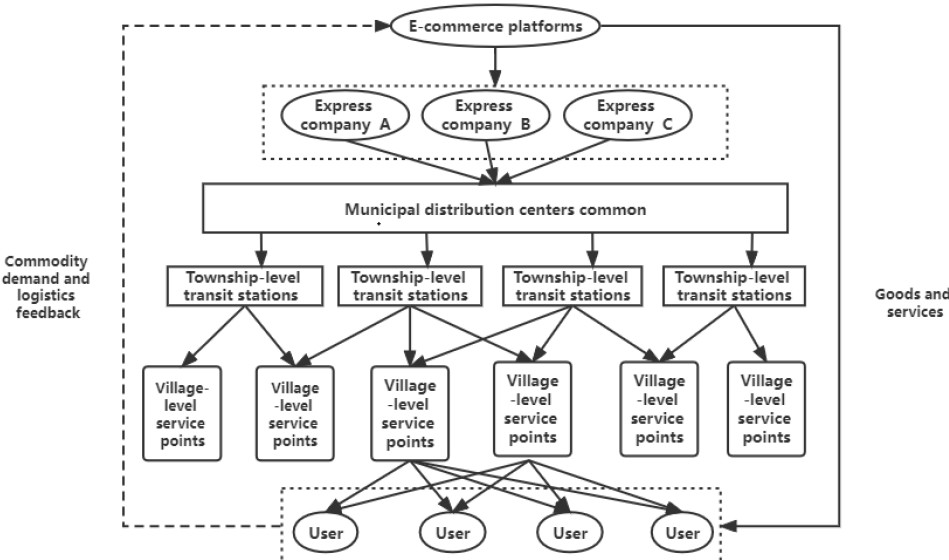

**Figure 1.** Schematic diagram of the three-level joint distribution system of rural "county-town-village".

As the source of the whole express delivery task, the e-commerce platform firstly displays products and provides goods and services through e-commerce platform, so as to meet the diversified consumer needs of users. Then, once the goods are sold, the e-commerce platform will entrust the goods to third-party logistics enterprises to complete the follow-up tasks [38,39]. The quality of express delivery will be the main factor to determine the sustainable development of e-commerce platforms, which is also the most concerning issue of consumers. Therefore, e-commerce platforms need to supervise and intervene in the distribution link.

County-level express joint distribution centers refer to the investment and construction of several express logistics companies supported and guided by the e-commerce platform [40,41]. The location of the joint distribution center should consider the transportation cost, market prospect and distribution of express companies. Under this model, county-level joint distribution mainly includes transportation management, a storage center and a comprehensive information management department. Specifically, the operation level of a county-level express joint distribution center, an important facility for commodity circulation, will directly determine the overall circulation efficiency and is the core of the success or failure of the three-level joint distribution system. The joint distribution center undertakes various functions such as goods collection, storage, sorting, circulation processing, delivery and information processing. For example, it collects logistics information of township transfer stations and village service points, summarizes the feedback information, and provides two-way information support for the distribution process. In contrast, in view of the complexity of the distribution process and the diversity of distribution needs, we give full play to the advantages of our own information platform, integrate all resources to achieve accurate and smooth logistics express transfer, and improve the quality of service.

Township-level transfer stations connect the preceding and the following in the whole rural express joint distribution system. It undertakes the goods of a county-level joint distribution center, carries out the secondary sorting of the goods and organizes and summarizes the village-level goods. The distributor sends the lower part to the next node [42,43].

Village service points are the end nodes of rural express delivery. Based on geographical location, road conditions, population distribution and other factors, the service scope of each station is scientifically constructed and divided. End service points generally include rural convenience store cooperation and company co-construction of self-pickup cabinets to facilitate residents' mail and pickup services [44–46].

### 2.2. Construction of an Evolutionary Game Model for Joint Distribution of Express Delivery Enterprises in Rural Areas

Joint distribution is an important means to solve the problem of express delivery in rural areas, but the research on the behavior of rural express enterprises under the competition and cooperation mechanism is insufficient. Therefore, an evolutionary game model can be established to analyze the cooperative competition behavior and its dynamic evolution process among different express delivery enterprises and to explore the influencing factors considered in the strategy selection of express delivery enterprises, the mutual competition and the cooperation mechanism [47].

#### 2.2.1. Basic Assumptions of the Evolutionary Game Model

(1)  Game player hypothesis

According to the intention of express companies to participate in joint distribution, express companies N will produce $2^n$ situations, which can be regarded as an evolutionary game between two factions of competition and cooperation regardless of the choice. When three express delivery companies participate in the alliance game, the final possible results are that none of the three cooperate, two of the three cooperate, and all three cooperate. When all three companies cooperate, their game can be interpreted as the game between any two cooperative alliances and the third company, which can still be regarded as a game between the two parties. Similarly, when four express companies participate in the game, the final possible results are that the four do not cooperate, any two cooperate, any

three cooperate or all four cooperate, and this process can be regarded as a game between the alliance that has been built and another alliance or company, and so on, and eventually can be regarded as the evolutionary game between the two sides. The two sides of the game may be a game between the small alliance and the small alliance, or a game between the small alliance and the express company, or a game between the express company and the express company. Therefore, in order to simplify the discussion of the model, go to the center of the problem and make the initial evolutionary game model accurate and conducive to calculation; this paper will therefore study the evolutionary game of both sides. Express companies M and N have bounded rationality, which is biased due to strategic misjudgment. Under the condition that complete information is not required, the game process of both sides is complicated, and the dynamic game equilibrium is reached through continuous trial and error [48].

(2)　　Behavioral strategy of the game

When the game is played, express companies M and N can only choose between competition or cooperation. Their strategy set is $S_n = \{cooperation, competition\}$, $n = (M, N)$. If express company M chooses cooperation probability $x(0 < x < 1)$, then competition probability is $1 - x$; if express company N chooses cooperation probability $y(0 < y < 1)$, then competition probability is $1 - y$ [49].

(3)　　Policy selection and parameters

The equation parameters are defined in Table 1.

**Table 1.** Algorithm parameters.

| Parameter | Meaning | |
|---|---|---|
| The operating income of the enterprise alone | $R_i$ i = M, N | |
| Excess returns from joint distribution | $\Delta R$ | |
| Excess return distribution coefficient | Excess companies M excess | |
| | Excess companies N excess benefit distribution factor $1 - \alpha$ | |
| Joint distribution of the initial input cost | $C_0$ | |
| Input cost apportionment coefficient | Express company M input cost sharing factor $\beta$ | |
| | Express company N input cost sharing factor $1 - \beta$ | |
| Joint distribution managing operating costs | $C_1$ | |
| Operating cost apportionment coefficient | Express company M operating | |
| | Express company N operating cos t sharing factor $1 - \gamma$ | |
| Risk-bearing coefficient of cooperative operation | Operating risk coefficient of express delivery enterprises sharing time f | |
| Default penalty cost | P | |
| Learning absorption capacity of enterprises | $L_i$ i = M, N | |
| Positive externality Benefits | $E_i$ i = M, N | |

**Hypothesis 1.** *The existence and evolution of excess returns is fundamental in the joint distribution system formed by express companies. Assume that the excess return in the joint distribution system during t is $\Delta R$, the distribution coefficient of the excess return of express company M is $\alpha$ $(0 < \alpha < 1)$, and the excess return is $\alpha \Delta R$; similarly, the distribution coefficient of enterprise N is $1 - \alpha$, and the excess return is $(1 - \alpha)\Delta R$.*

**Hypothesis 2.** *The implementation of joint distribution between express delivery enterprises can create positive effects for participants, namely, $\alpha \Delta R > L_M E_M, (1 - \alpha)\Delta R > L_N E_N$.*

**Hypothesis 3.** *The risk cost of cooperative operation is proportional to the total cost. Calculated by the proportional algorithm, the risk cost of cooperative operation is $(1 + f)(C_0 + C_1)$, and $f$ belongs to the interval $(0, 1)$.*

**Hypothesis 4.** *Both express company M and N are considered as finite rational economic agents. Therefore, for the default penalty cost to achieve the desired effect, it needs to meet $\beta C_0 + \gamma C_1 > P$ and $(1 - \beta)C_0 + (1 - \gamma)C_1 > P$; the input cost is greater than the penalty cost.*

Based on the above assumptions, the evolutionary income payment matrix between rural express joint distribution enterprises is shown in Table 2 [50,51]:

**Table 2.** Joint distribution between alliance enterprises evolution proceeds payoff matrix.

| Express Delivery Company M | Express Delivery Company N | |
|---|---|---|
| | Cooperation $y$ | Competition $1-y$ |
| Cooperation $x$ | $R_M + \alpha\Delta R - (1+f)(\beta C_0 + \gamma C_1)$, $R_N + (1-\alpha)\Delta R - (1+f)[(1-\beta)C_0 + (1-\gamma)C_1]$ | $R_M - (1+f)(\beta C_0 + \gamma C_1) + P$, $R_N + L_N E_N - P$ |
| Competition $1-x$ | $R_M + L_M E_M - P$, $R_N - (1+f)[(1-\beta)C_0 + (1-\gamma)C_1] + P$ | $R_M$ $R_N$ |

2.2.2. Evolutionary Game Model Solution

Let the expected return of express delivery company M be $U_M$, the expected return of cooperation be $U_{M1}$ and the expected return of competition be $U_{M2}$. The expected return of express delivery company N is $U_N$, the expected return of cooperation is $U_{N1}$ and the expected return of competition is $U_{N2}$ [52].

(1) Explore express delivery company M

Cooperation Strategy:

$$U_{M1} = y[R_M + \alpha\Delta R - (1+f)(\beta C_0 + \gamma C_1)] + (1-y)[R_M - (1+f)(\beta C_0 + \gamma C_1) + P]$$
$$= R_M + y\alpha\Delta R - (1+f)(\beta C_0 + \gamma C_1) + P - yP \tag{1}$$

The competitive strategy:

$$U_{M2} = y(R_M + L_M E_M - P) + (1-y)R_M$$
$$= R_M + yL_M E_M - yP \tag{2}$$

The average expected return of express company M in this income matrix is

$$\overline{U_M} = xU_{M1} + (1-x)U_{M2}$$
$$= R_M + xy(\alpha\Delta R - L_M E_M) + (x-y)P + yL_M E_M - x(1+f)(\beta C_0 + \gamma C_1) \tag{3}$$

The strategy selection of express delivery enterprise M in the process of the long-term repeated game replicates the dynamic equation:

$$F_M(x) = \frac{dx}{dt} = x(U_{M1} - \overline{U_M})$$
$$= x(1-x)[y(\alpha\Delta R - L_M E_M) + P - (1+f)(\beta C_0 + \gamma C_1)] \tag{4}$$

(2) Explore express delivery company N

Cooperation Strategy:

$$U_{N1} = x[R_N + (1-\alpha)\Delta R - (1+f)[(1-\beta)C_0 + (1-\gamma)C_1]] + (1-x)[[R_N - (1+f)[(1-\beta)C_0 + (1-\gamma)C_1] + P] \tag{5}$$

Competitive strategy:

$$U_{N2} = x(R_N + L_N E_N - P) + (1-x)R_N \tag{6}$$

The average expected return of express enterprise N in this income matrix is

$$\overline{U_N} = yU_{N1} + (1-y)U_{N2}$$
$$= R_N + xy[(1-\alpha)\Delta R - L_N E_N] + (y-x)P + xL_N E_N - y(1+f)[(1-\beta)C_0 + (1-\gamma)C_1] \tag{7}$$

The strategy selection of express delivery enterprise N in the long-term repeated game process replicates the dynamic equation:

$$F_N(y) = \frac{dy}{dt} = y(U_{N1} - \overline{U_N})$$
$$= y(1-y)[x[(1-\alpha)\Delta R - L_N E_N] + P - (1+f)[(1-\beta)C_0 + (1-\gamma)C_1] \tag{8}$$

At this time for M, N replication dynamic equation simplification, $C_M = (1+f)(\beta C_0 + \gamma C_1)C_N = (1+f)[(1-\beta)C_0 + (1-\gamma)C_1]$.

At this time, the two-dimensional continuous dynamic system of the strategic behavior of express companies M and N is:

$$\begin{cases} F_M(x) = x(U_{M1} - \overline{U_M}) \\ F_N(y) = y(U_{N1} - \overline{U_N}) \end{cases} \tag{9}$$

Which turns into:

$$\begin{cases} F_M(x) = x(1-x)[y(\alpha\Delta R - L_M E_M) + P - C_M] \\ F_N(y) = y(1-y)[x[(1-\alpha)\Delta R - L_N E_N] + P - C_N] \end{cases} \tag{10}$$

2.2.3. Stability Analysis of the Evolutionary Game Model

According to the basic definition of evolutionary game theory, both sides of the game achieve the stability of the game result by constantly adjusting their strategies and pursuing the final evolutionary stability strategy; namely, the requirements are $F(x) = 0$ and first-order derivative $F'(x) < 0$.

(1)  Analysis on the evolutionary stability of the selection strategy of express delivery company M

Let $F_M(x) = \frac{dx}{dt} = x(U_{M1} - \overline{U_M}) = 0$, solve

$$\begin{cases} x = 0 \\ x = 1 \\ y^* = \frac{C_M - P}{\alpha\Delta R - L_M E_M} \end{cases}$$

Find the first order derivative for $F_M(x)$:

$$F'(x) = \frac{\partial F(x)}{\partial x} = (1-2x)[y(\alpha\Delta R - L_M E_M) + P - C_M] \tag{11}$$

At the time $y = y^*$, $F(x) = 0$. When express company N chooses cooperation probability $y^*$, no matter what value express company M chooses for competition probability, the result is Nash equilibrium. Its phase diagram is shown in Figure 2a.

At the time $y \neq y^*$, according to the conditions given in hypothesis 4: $\beta C_0 + \gamma C_1 > P$, therefore $C_M - P > 0$, which is constant. Therefore, the size of $\alpha\Delta R - L_M E_M$ is explored and classified.

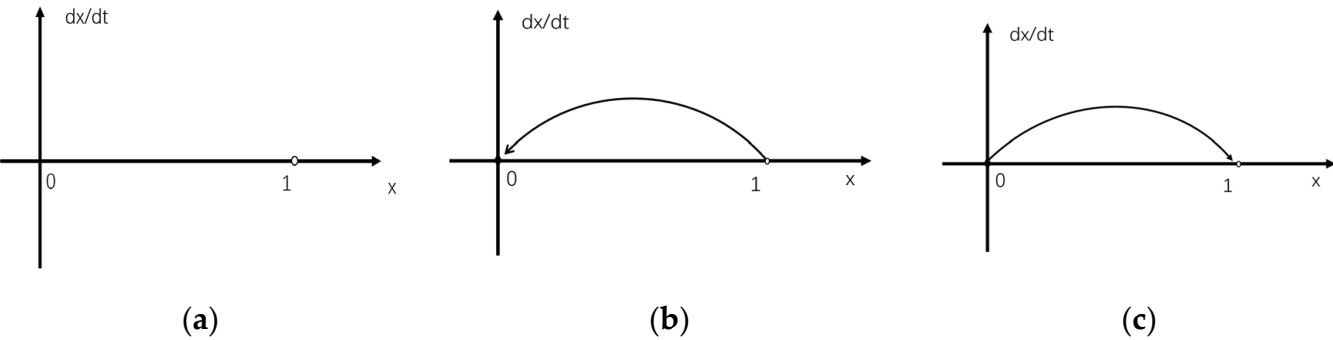

**Figure 2.** M logistics enterprise replicates the dynamic phase diagram under different values.

When $\alpha\Delta R - L_M E_M < 0$, $y > y^*$ holds; when $x = 0$, then $F'(x) < 0$; when $x = 1$, then $F'(x) > 0$. Therefore, according to the stability principle, $x = 0$ is a stable solution. In this case, when the value of excess earnings obtained by express company M after joint distribution is less than the betrayal cost of joint distribution, no matter what strategy express company N chooses, M only chooses the competitive strategy, and its phase diagram is shown in Figure 2b.

When $C_M - P > \alpha\Delta R - L_M E_M > 0$, $y^* > 1 > y$ still holds, as well as the solution of $F'(0)\langle 0, \ F'(1)\rangle 0$. Similarly, when the value of excess earnings obtained by express company M after joint distribution is less than the betrayal cost of joint distribution, no matter what strategy express company N chooses, M only chooses the competitive strategy, and its phase diagram is shown in Figure 2c.

When $C_M - P < \alpha\Delta R - L_M E_M$, it is necessary to make a classification discussion again.

When $y^* > y > 0$, then $F'(0)\langle 0, \ F'(1)\rangle 0$, the $x = 0$ is a stable solution, and the final result of the evolutionary game is that express company M chooses a competitive strategy. When $y > y^* > 0$, then $F'(0) > 0$, $F'(1) < 0$, the $x = 1$ is a stable solution, and the final result of the evolutionary game is that express company M only chooses the cooperation strategy.

According to the classification discussion, in order for express company M to participate in the joint distribution system, the prerequisite is that the value of excess earnings obtained by express company M after joint distribution is greater than the betrayal cost of the joint distribution. At this time, M's cooperation intention depends on Company N.

(2) Evolutionary stability analysis of the selection strategy of express delivery Company N.

Let $F_N(y) = \frac{dy}{dt} = y(U_{N1} - \overline{U_N}) = 0$, solve

$$\begin{cases} y = 0 \\ y = 1 \\ x^* = \frac{C_N - P}{(1-\alpha)\Delta R - L_N E_N} \end{cases}$$

Take the first derivative with respect to phi $F_N(y)$

$$F'(y) = \frac{\partial F(y)}{\partial y} = (1 - 2y)[x[(1 - \alpha)\Delta R - L_N E_N] + P - C_N] \tag{12}$$

At that time, $x = x^*$ and $F(y) = 0$; when express company N chooses cooperation probability $x^*$, no matter what value express company M chooses for competition probability, the result is the Nash equilibrium, and its phase diagram is shown in Figure 3a.

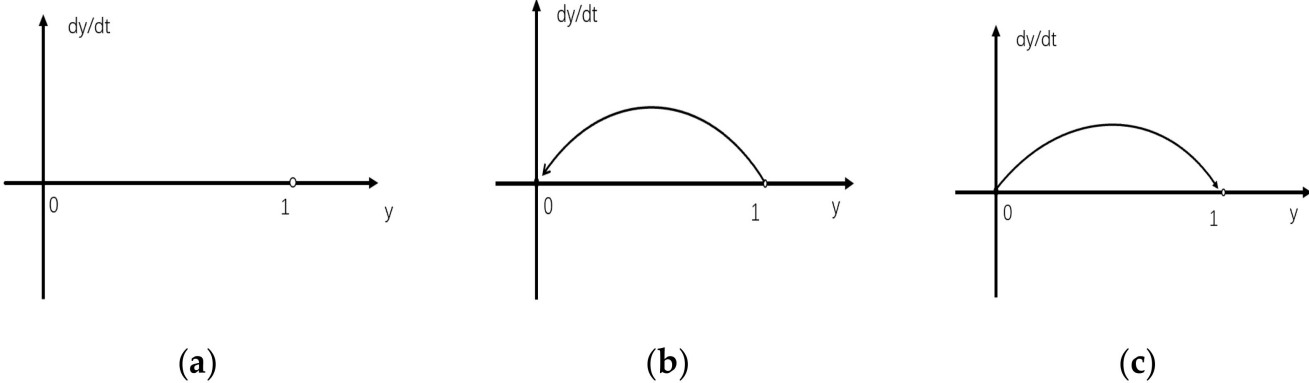

**Figure 3.** N logistics enterprise replicates the dynamic phase diagram under different values.

At that time, $x \neq x^*$, according to the conditions given in Hypothesis 4: $(1 - \beta)C_0 + (1 - \gamma)C_1 > P$; therefore, $C_N - P > 0$, and it is constant, so the size $(1 - \alpha)\Delta R - L_N E_N$ is explored and the classification is discussed.

When $(1 - \alpha)\Delta R - L_N E_N < 0$, at this time, $x > x^*$ holds; when $y = 0$, then $F'(y) < 0$; when $y = 1$, then $F'(y) > 0$. Therefore, according to the stability principle, $y = 0$ is a stable solution. In this case, when the value of excess earnings obtained by express company N after joint distribution is less than the betrayal cost of joint distribution, no matter what strategy is chosen by express company M, company N only chooses the competitive strategy, and its phase diagram is shown in Figure 3b.

When $C_N - P > (1 - \alpha)\Delta R - L_N E_N > 0$, at this time, $x^* > 1 > x$ holds, and the solution of $F'(0) \langle 0, F'(1) \rangle 0$. Similarly, when the value of excess earnings obtained by express company N after joint distribution is less than the betrayal cost of joint distribution, no matter what strategy is chosen by express company M, company N only chooses the competitive strategy, and its phase diagram is shown in Figure 3c.

When $C_N - P < (1 - \alpha)\Delta R - L_N E_N$, then you need to make a classification discussion.

When $x^* > x > 0$, then $F'(0) \langle 0 F'(1) \rangle 0$, the $y = 0$ is a stable solution, and the final result of the evolutionary game is that express company N chooses the competitive strategy. When $x > x^* > 0$, then $F'(0) > 0 F'(1) < 0$, the $y = 1$ is a stable solution, and the final result of evolutionary game is that express company N only chooses the cooperation strategy.

According to the classification discussion, in order for express company N to participate in the joint distribution system, the prerequisite is that the value of excess earnings obtained by express company N after joint distribution is greater than the betrayal cost of joint distribution. At this time, N's cooperation intention depends on Company M.

(3) Stability analysis of two-dimensional continuous dynamic system of express Company M and express Company N

According to the stability analysis of the two sides of the game, five local equilibrium points are obtained:

$$E_0(0,0) \; E_1(0,1) \; E_2(1,0) \; E_3(1,1) \; E_4 \left( \frac{C_N - P}{(1 - \alpha)\Delta R - L_N E_N}, \frac{C_M - P}{\alpha \Delta R - L_M E_M} \right)$$

The stability of local equilibrium points can be judged by using the method of the Yakerby matrix. The Accord matrix can be obtained by taking partial derivatives with respect to x and y respectively.

$$J = \begin{pmatrix} \frac{\partial F(x)}{\partial x} & \frac{\partial F(y)}{\partial y} \\ \frac{\partial F(y)}{\partial x} & \frac{\partial F(y)}{\partial y} \end{pmatrix}$$

Simplified to:

$$J = \begin{pmatrix} J_{11} & J_{12} \\ J_{21} & J_{22} \end{pmatrix}$$

Among them
$$
\begin{cases}
J_{11} = (1 - 2x)[y(\alpha\Delta R - L_M E_M) + P - C_M] \\
J_{12} = x(1 - x)(\alpha\Delta R - L_M E_M) \\
J_{21} = y(1 - y)[(1 - \alpha)\Delta R - L_N E_N] \\
J_{22} = (1 - 2y)[x[(1 - \alpha)\Delta R - L_N E_N] + P - C_N]
\end{cases}
$$

If the express company of M and N are at the end of the game system with the evolution of joint distribution accord, the determinant of a matrix is:

$$\det J = J_{11} \cdot J_{22} - J_{12} \cdot J_{21} \tag{13}$$

The trace of the Accord ratio matrix of the end-joint distribution evolutionary game system composed of express companies M and N is:

$$\text{tr}J = J_{11} + J_{22} \tag{14}$$

When the determinant of the Accord ratio matrix of the common distribution game system composed of express companies M and N is greater than 0—that is, $\det J > 0$—and the trace of the determinant is less than 0—that is, $\text{tr}J < 0$—the corresponding local equilibrium point is ESS. The stability of the five local equilibrium points can be judged by respectively substituting them into the formula. The results are shown in Table 3.

**Table 3.** Evolutionary game model of local stability point value.

| Equilibrium Point | detJ | Symbol | trJ | Symbol | Stability |
|---|---|---|---|---|---|
| $E_0(0,0)$ | $(P - C_N) * (P - C_M)$ | $+$ | $(P - C_N) + (P - C_M)$ | $-$ | ESS |
| $E_1(0,1)$ | $-(\alpha\Delta R - L_M E_M + P - C_M) * (P - C_N)$ | $+$ | $(\alpha\Delta R - L_M E_M + P - C_M) - (P - C_N)$ | indefinite | Be unstable |
| $E_2(1,0)$ | $-(P - C_M)*[(1 - \alpha)\Delta R - L_N E_N + P - C_N]$ | $+$ | $-(P - C_M) + [(1 - \alpha)\Delta R - L_N E_N + P - C_N]$ | $+$ | Be unstable |
| $E_3(1,1)$ | $[(\alpha\Delta R - L_M E_M) + P - C_M]* [(1 - \alpha)\Delta R - L_N E_N + P - C_N]$ | $+$ | $-[(\alpha\Delta R - L_M E_M) + P - C_M] - [(1 - \alpha)\Delta R - L_N E_N + P - C_N]$ | $-$ | ESS |
| $E_4(x^*, y^*)$ | $-x^* y^*(1 - x^*)(1 - y^*)* (\alpha\Delta R - L_M E_M)[(1 - \alpha)\Delta R - L_N E_N]$ | $-$ | $0$ | $0$ | Saddle point |

As can be seen from the table, the local equilibrium point $E_0(0,0)$ $E_3(1,1)$ is an evolutionarily stable strategy, that is, the final evolution result of the terminal joint distribution evolutionary game system composed of express companies M and N is either (cooperation, cooperation) or (competition, competition), thus both parties either choose the cooperation strategy or both parties choose the competition strategy. This indicates that one express company chooses the competitive strategy, and the other one will eventually also choose the competitive strategy. In order to achieve joint distribution, the two express companies need to choose the cooperation strategy at the same time. As can be seen from the table, the local equilibrium point $E_1(0,1)$ and $E_2(1,0)$ are the unstable, and $E_4(x^*, y^*)$ is saddle point. Therefore, the evolutionary game phase of the terminal joint distribution evolutionary game system composed of express company M and N is shown in Figure 4.

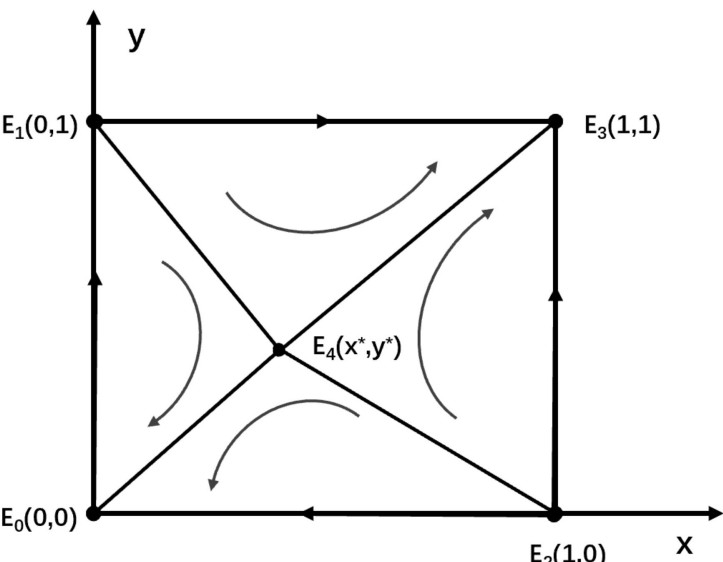

**Figure 4.** Phase diagram of an evolutionary game of joint distribution alliance.

According to the phase diagram of the evolutionary game, when the initial willingness of express companies M and N fall on the plane composed of $E_0(0,0)$, $E_1(0,1)$, $E_4(x^*, y^*)$, $E_2(1,0)$, the final result of the evolutionary game will be that both express companies choose a competitive strategy. When the initial willingness of express companies M and N falls on the plane composed of $E_1(0,1)$, $E_4(x^*, y^*)$, $E_2(1,0)$, $E_3(1,1)$, the final result of the evolutionary game will be that both express companies choose a cooperative strategy.

## 3. System Stability Analysis

### 3.1. Influence of Initial Willingness on the Evolutionary Game Model

Since the system is also sensitive to the initial state, a slight change in the initial state will change the convergence direction and stable state of the whole system. When the initial willingness of express delivery enterprises on both sides of the game falls within the quadrilateral $S_1(E_0 E_1 E_4 E_2)$, according to the phase diagram of the evolutionary game, the final trend point $E_0(0,0)$ of both sides of the game is that both sides choose the competitive strategy. When the initial willingness of express enterprises on both sides of the game falls within the quadrilateral $S_2(E_3 E_1 E_4 E_2)$, it can be seen from the phase diagram of the evolutionary game that the two sides of the game eventually tend toward the point $E_3(1,1)$, thus both sides choose the cooperation strategy.

### 3.2. The Influence of Parameters on the Evolutionary Game Model

According to the phase diagram of the evolutionary game, the evolutionary path of the joint distribution system depends on the quadrilateral $S_1$ and the area of the quadrilateral $S_2$. When $S_1 > S_2$, the probability of the express company deviating from the alliance is greater than the probability of cooperation; when $S_1 < S_2$, the probability of the express company deviating from the alliance is less than the probability of cooperation. Therefore, through the $S_2$ change of area, the influencing factors of alliance cooperation can be obtained, and its area is:

$$S_1 = \frac{1}{2}\left(\frac{C_N - P}{(1-\alpha)\Delta R - L_N E_N} + \frac{C_M - P}{\alpha\Delta R - L_M E_M}\right) \tag{15}$$

$$S_2 = 1 - S_1$$

$$
\begin{aligned}
S_2 &= 1 - \frac{1}{2}\left(\frac{C_N - P}{(1-\alpha)\Delta R - L_N E_N} + \frac{C_M - P}{\alpha\Delta R - L_M E_M}\right) \\
&= 1 - \frac{1}{2}\left(\frac{(1+f)[(1-\beta)C_0 + (1-\gamma)C_1] - P}{(1-\alpha)\Delta R - L_N E_N} + \frac{(1+f)(\beta C_0 + \gamma C_1) - P}{\alpha\Delta R - L_M E_M}\right)
\end{aligned} \tag{16}
$$

It can be seen from the formula that factors affecting the evolutionary game model mainly include excess return $\Delta R$, excess return distribution coefficient $\alpha$, initial input cost $C_0$, initial input cost allocation factor $\beta$, operating cost $C_1$, operating cost allocation factor $\gamma$, cooperative risk-bearing coefficient f, penalty cost for breach of contract P, learning and absorption capacity of the firm $L_i$ and positive external benefit of the firm $E_i$.

### 3.2.1. The Influence of Excess Returns on Game Results

Take the partial derivative of the influencing factor:

$$\frac{\partial_{S_2}}{\partial_{\Delta R}} = \frac{1}{2}\left( \frac{(C_N - P)(1 - \alpha)}{[(1 - \alpha)\Delta R - L_N E_N]^2} + \frac{(C_M - P)\alpha}{(\alpha \Delta R - L_M E_M)^2} \right) > 0$$

Right $S_2$ is about the excess return $\Delta R$ of the monotonically increasing function. Therefore, the higher the initial input cost, the larger the polygon $S_2$ area and the greater the probability of cooperation. Both sides of the game face a large number of excess returns to stimulate the enthusiasm of the participants in cooperation.

### 3.2.2. The Influence of Excess Return Distribution Coefficient on Game Results

Take the partial derivative of the influencing factor:

$$\frac{\partial_{S_2}}{\partial_{\alpha}} = -\frac{1}{2}\left( \frac{(C_N - P)\Delta R}{[(1 - \alpha)\Delta R - L_N E_N]^2} - \frac{(C_M - P)\Delta R}{[\alpha \Delta R - L_M E_M]^2} \right)$$

The first derivative of the excess return distribution coefficient $\alpha$ is solved, but the magnitude cannot be determined, so the second derivative $\alpha$ is solved.

$$\frac{\partial^2_{S_2}}{\partial^2_{\alpha}} = -\frac{(C_N - P)\Delta R^2}{[(1 - \alpha)\Delta R - L_N E_N]^3} - \frac{(C_M - P)\Delta R^2}{[\alpha \Delta R - L_M E_M]^3} < 0$$

According to the definition of the second derivative, it can be seen that the formula $S_2$ is a convex function with a maximum value, which presents the trend of first increasing and then decreasing. When $\frac{\partial_{S_2}}{\partial_{\alpha}} = -\frac{1}{2}\left( \frac{(C_N - P)\Delta R}{[(1 - \alpha)\Delta R - L_N E_N]^2} - \frac{(C_M - P)\Delta R}{[\alpha \Delta R - L_M E_M]^2} \right) = 0$, the maximum value is obtained immediately. When other parameters remain unchanged, there is a number of excess return distribution coefficients that make both game parties have a suitable return scheme, i.e., $\frac{(C_N - P)\Delta R}{[(1 - \alpha)\Delta R - L_N E_N]^2} = \frac{(C_M - P)\Delta R}{[\alpha \Delta R - L_M E_M]^2}$.

### 3.2.3. The Influence of Initial Input Cost on Game Outcome

Take the partial derivative of the influencing factor:

$$\frac{\partial_{S_2}}{\partial_{C_0}} = -\frac{1}{2}\left( \frac{(1 + f)(1 - \beta)}{(1 - \alpha)\Delta R - L_N E_N} + \frac{(1 + f)\beta}{\alpha \Delta R - L_M E_M} \right) < 0$$

Right $S_2$ is about the upfront input cost input $C_0$ of monotonically decreasing function. Therefore, the higher the early input cost, the smaller the polygon $S_2$ area and the smaller the probability of cooperation. In a practical sense, excessive investment in the early stage of joint distribution alliance construction and worries about the uncertainty of future earnings will lead to the deviation of cooperation willingness of game enterprises.

### 3.2.4. The Influence of Input Cost Apportionment Coefficient on Game Result

Take the partial derivative of the influencing factor:

$$\frac{\partial_{S_2}}{\partial_{\beta}} = -\frac{1}{2}\left( \frac{(1 + f)C_0}{\alpha \Delta R - L_M E_M} - \frac{(1 + f)C_0}{(1 - \alpha)\Delta R - L_N E_N} \right)$$

It is necessary to make an assessment of the $\alpha\Delta R - L_M E_M$ with $(1-\alpha)\Delta R - L_N E_N$ size.

When $\alpha\Delta R - L_M E_M > (1-\alpha)\Delta R - L_N E_N$, then $\frac{\partial_{S_2}}{\partial_\beta} > 0$, the $S_2$ is about the monotonically increasing function of cost apportionment coefficient, the $\beta$ is larger, the $S_2$ area is larger, and the greater the probability of comprehensive cooperation. This formula means that when the difference between the excess returns and the external economic returns obtained by the non-cooperation strategy of express delivery enterprise M is higher than that of express delivery enterprise N, express delivery enterprise M needs to bear more costs.

When $\alpha\Delta R - L_M E_M < (1-\alpha)\Delta R - L_N E_N$, then $\frac{\partial_{S_2}}{\partial_\beta} < 0$, the $S_2$ is a monotone decreasing function of cost apportionment coefficient, the $\beta$ is larger, the $S_2$ area is smaller, and the smaller the probability of comprehensive cooperation. This formula means that when the difference between the excess returns of cooperation and the external economic returns obtained by the non-cooperation strategy of express delivery enterprise N is higher than that of express delivery enterprise M, express delivery enterprise N needs to bear more costs.

### 3.2.5. The Influence of Managing Operation Cost on Game Result

Take the partial derivative of the influencing factor:

$$\frac{\partial_{S_2}}{\partial_{C_1}} = -\frac{1}{2}\left(\frac{(1+f)(1-\gamma)}{(1-\alpha)\Delta R - L_N E_N} + \frac{(1+f)\gamma}{\alpha\Delta R - L_M E_M}\right) < 0$$

Operating cost $C_1$ is not the influence of initial conditions on evolutionary game, but $S_2$ is a monotone decreasing function of operating cost $C_1$. Therefore, the higher the operating cost $C_1$, the smaller the polygon $S_2$ area, and the smaller the probability of choosing cooperation. In a practical sense, the cost of maintaining the operation of the joint distribution alliance increases, leading to the situation that the enterprise cannot make a profit, which leads to the failure of the game parties during the construction of the evolutionary game.

### 3.2.6. The Influence of Operation Cost Apportionment Coefficient on Game Result

Take the partial derivative of the influencing factor:

$$\frac{\partial_{S_2}}{\partial_\gamma} = -\frac{1}{2}\left(\frac{(1+f)C_1}{\alpha\Delta R - L_M E_M} - \frac{(1+f)C_1}{(1-\alpha)\Delta R - L_N E_N}\right)$$

It is necessary to make an assessment of the $\alpha\Delta R - L_M E_M$ with $(1-\alpha)\Delta R - L_N E_N$ size.

(1) When $\alpha\Delta R - L_M E_M > (1-\alpha)\Delta R - L_N E_N$, then $\frac{\partial_{S_2}}{\partial_\gamma} > 0$, the $S_2$ is a monotonically increasing function of the cost apportionment coefficient. The $\gamma$ is larger, the $S_2$ area is larger and there is a greater probability of comprehensive cooperation. This formula means that when the difference between the excess returns and the external economic returns obtained by the non-cooperation strategy of express delivery enterprise M is higher than that of express delivery enterprise N, express delivery enterprise M needs to bear more costs.

(2) When $\alpha\Delta R - L_M E_M < (1-\alpha)\Delta R - L_N E_N$, then $\frac{\partial_{S_2}}{\partial_\gamma} < 0$, the $S_2$ is a monotone decreasing function of cost apportionment coefficient, the $\gamma$ is larger, the $S_2$ area is smaller and the probability of comprehensive cooperation is smaller. This formula means that when the difference between the excess returns of cooperation and the external economic returns are obtained by non-cooperation strategies, express delivery enterprise N is higher than that of express delivery enterprise M, thus express delivery enterprise N needs to bear more costs.

### 3.2.7. The Influence of the Cooperation Risk-Bearing Coefficient on Game Result

Take the partial derivative of the influencing factor:

$$\frac{\partial_{S_2}}{\partial_f} = -\frac{1}{2}\left(\frac{(1-\beta)C_0 + (1-\gamma)C_1}{[(1-\alpha)\Delta R - L_N E_N]^2} + \frac{(\beta C_0 + \gamma C_1)}{(\alpha\Delta R - L_M E_M)^2}\right) < 0$$

Thus, it can be seen that $S_2$ is a monotone decreasing function of cooperation risk taking coefficient f. Therefore, the greater the cooperation risk, the lower the probability of both parties choosing the cooperation strategy. In a practical sense, when an enterprise is faced with an unfavorable choice, it often takes evasive measures.

### 3.2.8. The Influence of Default Penalty Cost on Game Result

Take the partial derivative of the influencing factor:

$$\frac{\partial_{S_2}}{\partial_P} = \frac{1}{2}\left(\frac{1}{(1-\alpha)\Delta R - L_N E_N} + \frac{1}{\alpha\Delta R - L_M E_M}\right) > 0$$

It can be seen that it is a monotone increasing function of penalty cost for breach of contract. Therefore, the higher the penalty cost for breach of contract is, the higher the probability of both parties in the game choosing the cooperation strategy is $S_2P$. Because the high penalty cost is a constraint to both sides of the game, it can achieve the stability of the joint distribution alliance.

### 3.2.9. The Influence of Enterprises' Learning and Absorption Capacity on Game Results

Take the partial derivative of the influencing factor:

$$\frac{\partial_{S_2}}{\partial_{L_M}} = -\frac{1}{2}\left(\frac{(C_M - P)E_M}{(\alpha\Delta R - L_M E_M)^2}\right) < 0$$

Take enterprise M as an example to prove, and enterprise N is the same. Thus, it can be seen that $S_2$ is a monotone decreasing function about the learning and absorption capacity $L_i$ of the enterprise. Therefore, the higher the learning and absorption capacity of the enterprise, the lower the probability that the two sides of the game choose the cooperation strategy. Both parties in the game have strong substitution, and when one partner has strong resource capacity and absorption capacity, it is very likely to merge the enterprise with weak resource capacity, so as to monopolize the market share. More attention should be paid to the prevention of such risks when establishing the distribution alliance, so as to ensure the cooperation rights and interests of both parties.

### 3.2.10. The Influence of Positive External Benefits on Evolution Results

Take the partial derivative of the influencing factor:

$$\frac{\partial_{S_2}}{\partial_{E_M}} = -\frac{1}{2}\left(\frac{(C_M - P)L_M}{(\alpha\Delta R - L_M E_M)^2}\right) < 0$$

Take enterprise M as an example to prove, and enterprise N is the same. It can be seen that $S_2$ is a monotone decreasing function about the positive external benefits $E_M$ of the enterprise. Therefore, the higher the learning and absorption capacity of the enterprise, the lower the probability of the two game parties choosing the cooperation strategy. If one of the players of the game economy can obtain its own external economic benefits, the system tends to converge to the non-cooperative strategy state.

### 3.3. Introduce the Evolutionary Game Model of Rewards and Punishments of e-Commerce Platforms

Based on the above analysis of the joint distribution evolutionary game model among express companies, where only express companies participate, it can be seen that without the guidance of an e-commerce platform, the results of the final evolutionary game between express companies will converge to two points: $E_0(0,0)$, $E_3(1,1)$. This is because the express companies involved in the game are all bounded rationally, and the ultimate goal is to maximize their own benefits. Therefore, the final result of the evolutionary game of joint distribution between rural express companies may be that both sides adopt competitive strategies, which ultimately makes the whole game process fall into the prisoner's dilemma. Therefore, it is necessary for e-commerce platforms to participate in the guidance and formulate reasonable reward and punishment mechanisms to promote express companies to carry out terminal joint distribution. when express companies adopt cooperative strategies, the e-commerce platform will give certain subsidies to encourage them; When they choose a competitive strategy, the e-commerce platform will charge certain fines to them, thus guiding the express company to implement terminal joint distribution.

#### 3.3.1. Payment Matrix Construction

In the evolutionary game model of terminal joint distribution among express companies, which introduces the reward and punishment factors of e-commerce platforms, the platform reward is $T_M$ and $T_N$; when the express company conducts joint distribution, the platform will provide financial subsidies in certain ways. Platform punishment is $F_M$ and $F_N$. The platform punishment is divided into two situations: one is that express companies that do not participate in the joint distribution need to pay a fine, the other is that express companies that participate in the joint distribution and then withdraw from the joint distribution need to pay a fine. The income payment matrix is shown in Table 4.

**Table 4.** Introduction of the evolution of the rewards and punishment factor after earnings payoff matrix.

| Express Delivery Company M | Express Delivery Company N | |
|---|---|---|
| | Cooperation for $y$ | Competition $1-y$ |
| Cooperation x | $R_M + T_M + \alpha\Delta R - (1+f)(\beta C_0 + \gamma C_1)$, $R_N + T_N + (1-\alpha)\Delta R - (1+f)[(1-\beta)C_0 + (1-\gamma)C_1]$ | $R_M + T_M - (1+f)(\beta C_0 + \gamma C_1) + P$, $R_N - F_N + L_N E_N - P$ |
| Competition $1-x$ | $R_M - F_M + L_M E_M - P$, $R_N + T_N - (1+f)[(1-\beta)C_0 + (1-\gamma)C_1] + P$ | $R_M - F_M$ $R_N - F_N$ |

#### 3.3.2. Model Solving

(1)　When exploring express company M:

Cooperation Strategy:

$$U_{M1} = y[R_M + T_M + \alpha\Delta R - (1+f)(\beta C_0 + \gamma C_1)] + (1-y)[R_M + T_M - (1+f)(\beta C_0 + \gamma C_1) + P] \tag{17}$$

The Competitive strategy:

$$U_{M2} = y(R_M + T_N + L_M E_M - P) + (1-y)(R_M - F_M) \tag{18}$$

The average expected return of express company M in this income matrix is

$$\overline{U_M} = xU_{M1} + (1-x)U_{M2}$$
$$= R_M - F_M + x[T_M + F_M + P - (1+f)(\beta C_0 + \gamma C_1)] + y(T_N + F_M + L_M E_M - P) + xy(\alpha\Delta R - T_N - F_M - L_M E_M) \tag{19}$$

The strategy selection of express delivery enterprise M in the process of long-term repeated game replicates the dynamic equation:

$$F_M(x) = \frac{dx}{dt} = x(U_{M1} - \overline{U_M})$$
$$= x(1-x)[y(\alpha\Delta R - T_N - F_M - L_M E_M) + T_M + F_M + P - (1+f)(\beta C_0 + \gamma C_1)] \tag{20}$$

(2)    When exploring express delivery company N:

Cooperation Strategy:

$$U_{N1} = x[R_N + T_N + (1-\alpha)\Delta R - (1+f)[(1-\beta)C_0 + (1-\gamma)C_1]] +$$
$$(1-x)[R_N + T_N - (1+f)[(1-\beta)C_0 + (1-\gamma)C_1] + P] \tag{21}$$

Competitive strategy:

$$U_{N2} = x(R_N - F_N + L_N E_N - P) + (1-x)(R_N - F_N) \tag{22}$$

The average expected return of express enterprise N in this income matrix is:

$$\overline{U_N} = yU_{N1} + (1-y)U_{N2}$$
$$= R_N - F_N + xy[(1-\alpha)\Delta R - L_N E_N] + (y-x)P + xL_N E_N + y[F_N + T_N - (1+f)[(1-\beta)C_0 + (1-\gamma)C_1] \tag{23}$$

The strategy selection of express delivery enterprise N in the long-term repeated game process replicates the dynamic equation:

$$F_N(y) = \frac{dy}{dt} = y(U_{N1} - \overline{U_N})$$
$$= y(1-y)[x[(1-\alpha)\Delta R - L_N E_N] + F_N + T_N + P - (1+f)[(1-\beta)C_0 + (1-\gamma)C_1] \tag{24}$$

At this time for M, the N replication dynamic equation simplification is:

$$C_M = (1+f)(\beta C_0 + \gamma C_1) \quad C_N = (1+f)[(1-\beta)C_0 + (1-\gamma)C_1]$$

At this time, the two-dimensional continuous dynamic system of the strategic behavior of express companies M and N is:

$$\begin{cases} F_M(x) = x(U_{M1} - \overline{U_M}) \\ F_N(y) = y(U_{N1} - \overline{U_N}) \end{cases} \tag{25}$$

Which turns into:

$$\begin{cases} F_M(x) = x(1-x)[y(\alpha\Delta R - T_N - F_M - L_M E_M) + T_M + F_M + P - C_M] \\ F_N(y) = y(1-y)[x[(1-\alpha)\Delta R - L_N E_N] + F_N + T_N + P - C_N] \end{cases} \tag{26}$$

According to the definition of Ess in evolutionary game theory, to obtain the local equilibrium point of the evolutionary game, $F(x) = 0$ and the first derivative $F\prime(x) < 0$.

Solving the game theory of the five new stable points $E_0(0,0)$, $E_1(0,1)$, $E_2(1,0)$, $E_3(1,1)$, $E_4'(x^*, y^*)$, the new points coordinates are $E_4'\left(\frac{C_N - P - F_N - T_N}{(1-\alpha)\Delta R - L_N E_N}, \frac{C_M - P - T_M - F_M}{\alpha\Delta R - L_M E_M}\right)$. Comparing with the original model, the following conclusions are drawn:

$$\frac{C_N - P - F_N - T_N}{(1-\alpha)\Delta R - L_N E_N} < \frac{C_N - P}{(1-\alpha)\Delta R - L_N E_N}, \frac{C_M - P - T_M - F_M}{\alpha\Delta R - L_M E_M} < \frac{C_M - P}{\alpha\Delta R - L_M E_M}$$

At this point, according to the analysis in the previous section, it is known that when the reward and punishment factors are incorporated into the model, the probability of the express company finally choosing the cooperation strategy will become larger. The evolutionary phase diagram of the e-commerce platform after the introduction of rewards and punishments is shown in Figure 5, and the probability of cooperation of the game subject increases by $S\left(E_4'E_1E_4E_2\right)$. It is thus concluded that when the platform adopts

a suitable reward and punishment mechanism, it will effectively promote the end joint distribution among the express company.

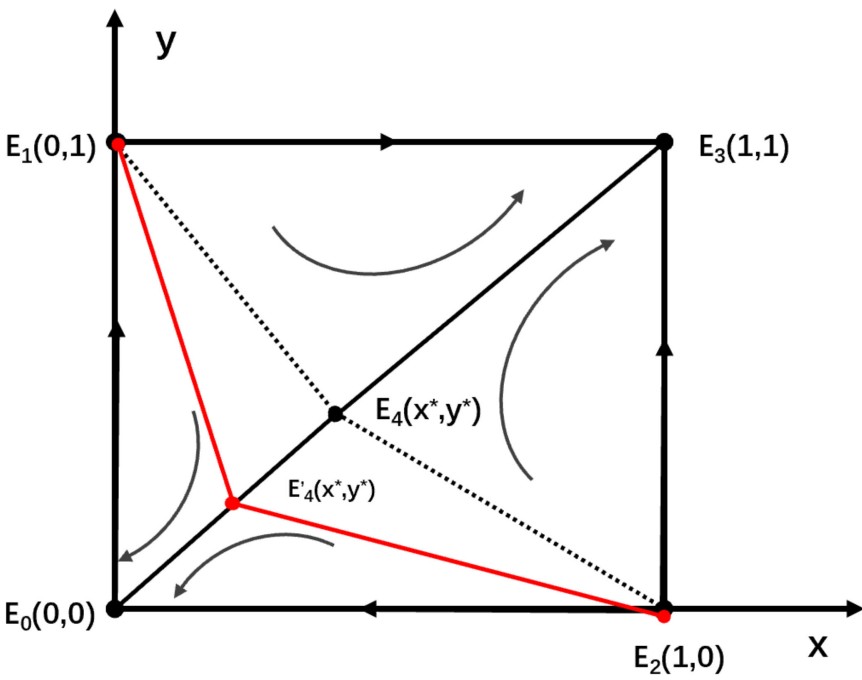

**Figure 5.** Platform phase diagram rewards and punishments after introduction of the evolution of the game.

## 4. Numerical Analysis

### 4.1. Parameter Assignment of Evolutionary Game Model

It is assumed that there are express companies M and N, and according to the parameter hypothesis of the evolutionary game model for the joint distribution of rural express companies in the two-dimensional continuous dynamic System—Section 2.2.3, each variable is assigned according to the conditions.

The values of the parameters in the game model are set as follows: the excess return is $\Delta R = 100$, the initial input cost is $C_0 = 30$, the managing operating cost is $C_1 = 20$, the excess return distribution coefficient, the initial input cost-bearing coefficient and the managing operating cost distribution coefficient are 0.55, the risk-bearing coefficient is $f = 0.3$, the learning ability of express enterprise M is 1.5, the learning ability of express enterprise N is 1.2, the positive external benefit that express enterprise M can obtain is 20 and the positive external benefit of enterprise N is 15. The agreed penalty is 15. The specific values of related parameters are shown in Table 5.

**Table 5.** Parameter assignment table.

| Influencing Factors | $\alpha$ | $\Delta R$ | $L_M$ | $E_M$ | $P$ | $L_N$ |
|---|---|---|---|---|---|---|
| The numerical | 0.55 | 100 | 1.5 | 20 | 15 | 1.3 |
| Influencing factors | $E_N$ | f | $\beta$ | $C_0$ | $\gamma$ | $C_1$ |
| The numerical | 18 | 0.3 | 0.55 | 25 | 0.55 | 15 |

### 4.2. Numerical Simulation Analysis of Influencing Factors of Evolutionary Game Model

#### 4.2.1. Influence of Initial Cooperation Intention on System Convergence Direction

In order to select the representative probability value of initial willingness to cooperate, we put the above assigned value into the replication dynamic equation and obtained the threshold value of approximately $E_4(0.389, 0.544)$. We select the values near the broken

line formed by point $E_1(0, 1)$ and point $E_2(1, 0)$, respectively, in order to better reflect the sensitivity of the system to the initial state. The initial willingness to cooperate is selected as the following four groups of values. The comparison of system simulation operation results is shown in Figure 6.

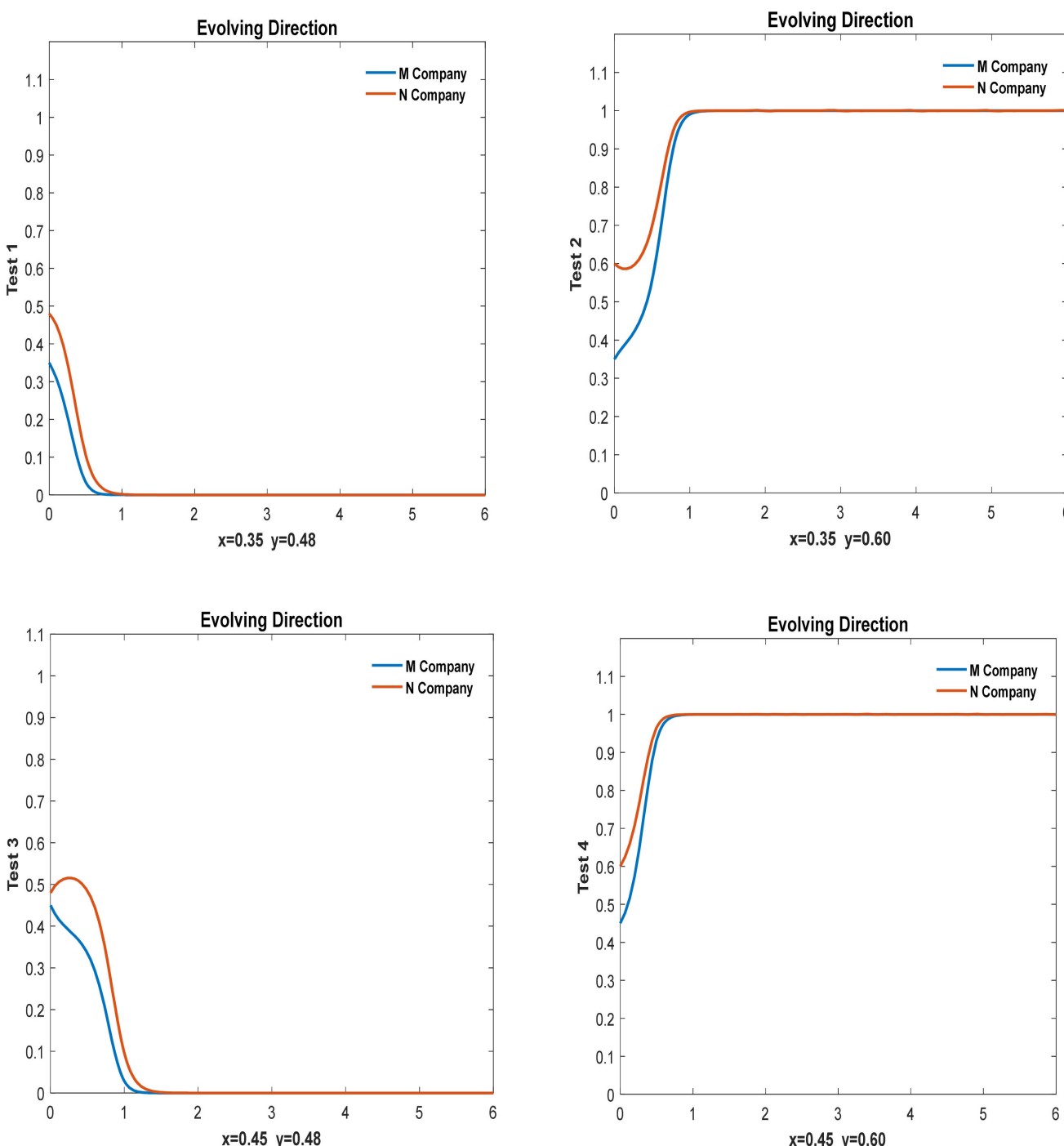

**Figure 6.** Influence of initial cooperation intention on system evolution results.

The initial willingness to cooperate takes the value 1: x = 0.35 y = 0.48.
The initial willingness to cooperate takes the value 2: x = 0.35 y = 0.60.
The initial willingness to cooperate takes the value 3: x = 0.45 y = 0.48.
The initial willingness to cooperate takes the value 4: x = 0.45 y = 0.60.

### 4.2.2. Influence of Excess Return and Its Distribution Coefficient

(1)    The influence of excess returns on the evolutionary game model

In order to more directly observe the influence of excess return on the evolution result, first, on the basis of the analysis of the impact of initial willingness to cooperate on the evolutionary results, we set the initial willingness to cooperate x = 0.45 y = 0.48. At this time, the system evolution result converges to 0 and the system is in a state of full competition. On this basis, the initial allocation of excess return of 100 is adjusted slightly upward and downward. The following simulation results are obtained by taking 105 and 95.

From the result Figure 7, it can be seen that when x = 0.45 y = 0.48, the system is set on the basis of the assignment, the initial willingness to cooperate is located $E_1E_4E_2$ below the fold. At this time, the evolutionary result of the system is non-cooperative between the two sides of the game. When other conditions are certain, a small increase or decrease in the excess return will prompt a shift in the evolutionary result of the system. The excess return increases by 5 units, the evolutionary path of the system undergoes a qualitative change and the evolution stabilizes in full cooperation. Conversely, the cooperation return decreases make the system tend toward the non-cooperative strategy faster. This further justifies the analytical results of the sensitivity of the system evolutionary outcome to excess returns. When the total excess returns increase, the probability of cooperation between the two sides of the game increases. When the excess returns reach a certain limit or more, the game outcome will eventually evolve toward full cooperation. When the excess returns decrease, the probability of non-cooperation between the two sides of the game increases. When it decreases below a certain limit, the game outcome will evolve toward a non-cooperative strategy.

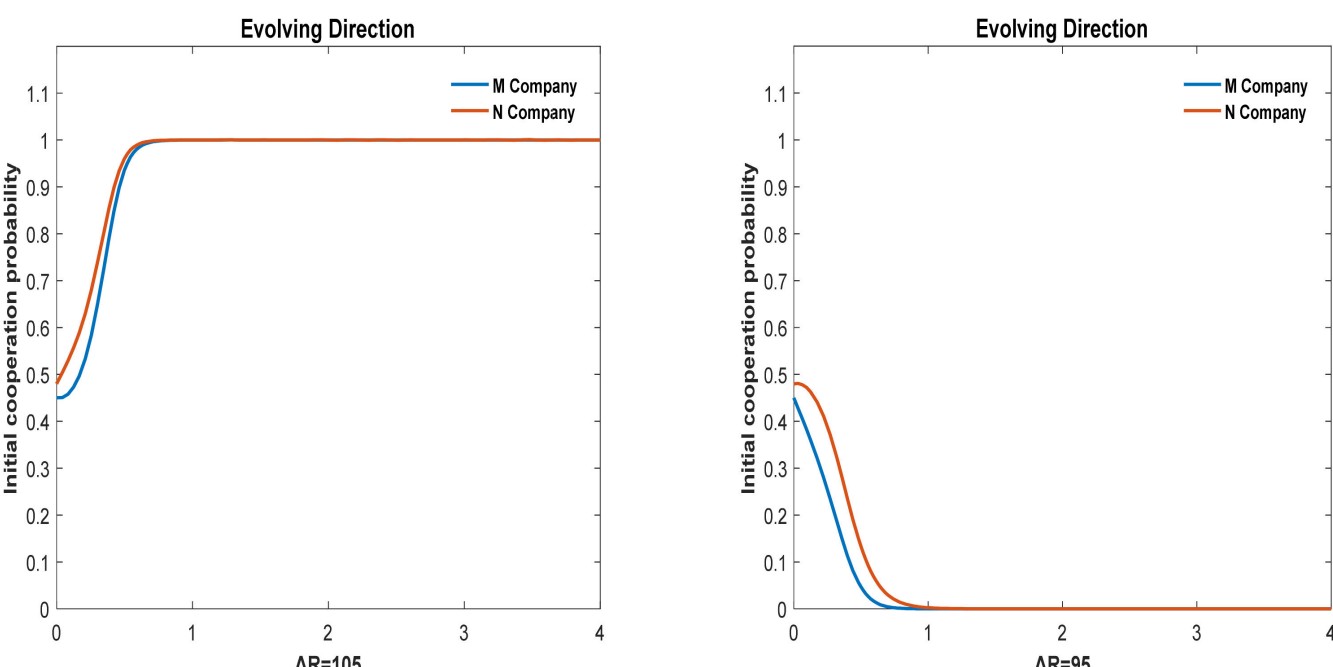

**Figure 7.** Excess returns for the influence of system evolution results.

(2)    The influence of excess return distribution coefficient on the evolutionary game model

When other parameters are consistent as above, select x = 0.45 y = 0.60. When $\alpha = 0.55$, the express delivery enterprise M bears 55% of the initial construction cost and 55% of the later input cost. At this time, the initial state of the game is above the broken line $E_1E_4E_2$, and the final result of the game is that both parties reach a comprehensive cooperation strategy. When express delivery company M bears the same cost, the income distribution coefficient decreases from 55% to 45%, and its willingness to cooperate

decreases. The evolution result eventually changes from comprehensive cooperation to non-cooperation. The evolution trend is shown in Figure 8.

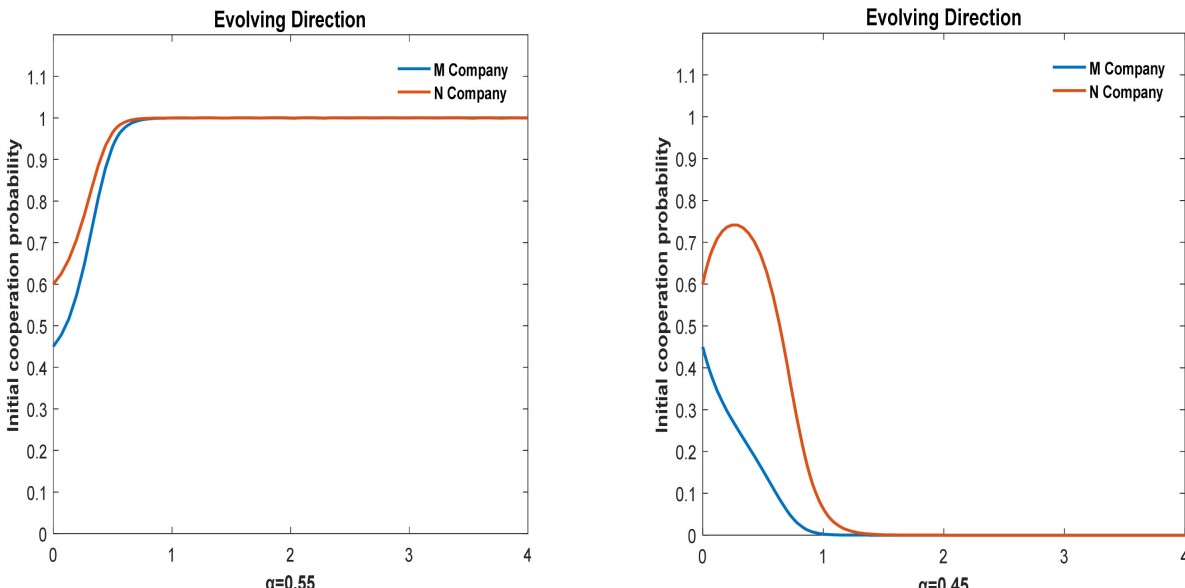

**Figure 8.** Excess return distribution coefficient for the influence of system evolution results.

### 4.2.3. Influence of Initial Input Cost and Its Distribution Coefficient

(1)  The impact of initial input cost on the evolutionary game model

According to the evolutionary simulation results in Figure 9, when the total initial construction cost is 25 and the initial cooperation willingness of the game parties is x = 0.45 y = 0.60, the system will eventually converge and stabilize to the point of full cooperation. However, when other conditions remain unchanged, the initial input cost of cooperation increases to 35 and the system evolution eventually tends toward non-cooperation. It can be proven that the increase of initial input cost of cooperation leads to the increase of non-cooperation willingness of both parties. When the increase reaches a certain limit, the evolutionary path of the system will develop qualitative changes. Ultimately, it tends to be non-cooperative.

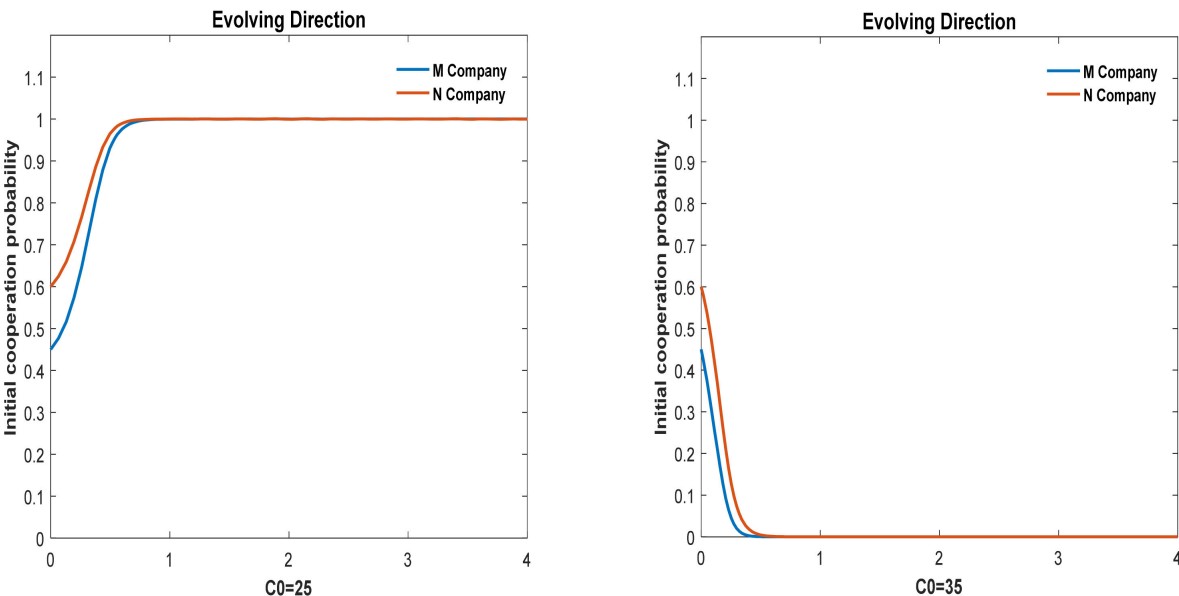

**Figure 9.** Influence on the result of the initial investment cost for system evolution.

(2)   The influence of initial investment coefficient on the evolutionary game model

When the initial cooperation willingness of both parties in the game is x = 0.45 y = 0.60 and the other conditions remain unchanged, it can be seen from Figure 10 that when the third-party logistics enterprise M enjoys 55% of the excess revenue and bears 55% of the initial cost, the two sides can reach a consensus on the cooperation of joint distribution. When the third-party logistics company M enjoys 55% excess returns but reduces the share of cost borne, the third-party logistics company N is less willing to cooperate, and it cannot accept the treatment that the third-party logistics company M bears the same costs but does not enjoy the same benefits, and the cooperation will eventually be difficult to maintain and will lead to a complete breakdown.

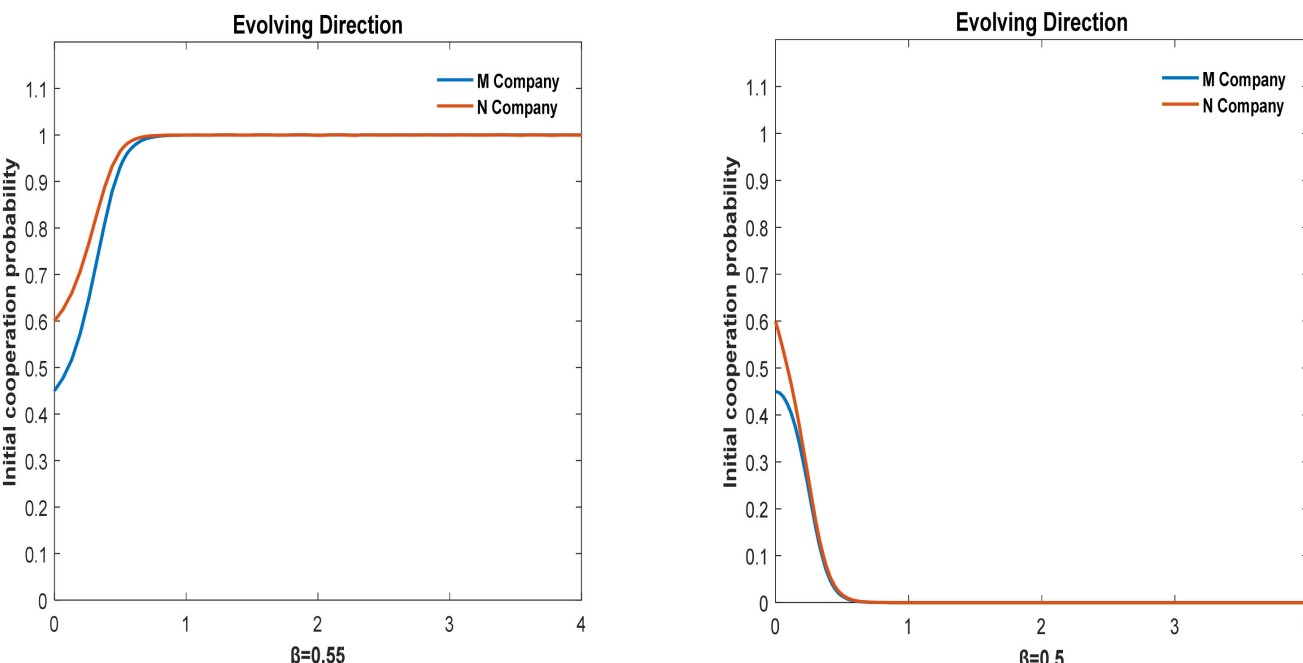

**Figure 10.** Coefficient of initial investment cost for the influence of system evolution results.

4.2.4. Influence of Managing Operating Cost and Its Distribution Coefficient

(1)   The influence of operational costs on the evolutionary game model

When the initial willingness of the two parties to cooperate is x = 0.45 y = 0.60 and the managing operating cost is 15, it can be seen from the evolutionary simulation results in Figure 11 that the system will eventually converge and stabilize to the point of full cooperation. However, when other conditions remain unchanged and the operating cost increases to 20, the system evolution at this time will eventually tend toward non-cooperation. The conclusion is that the increase of operational costs in the later stage of cooperation increases the willingness of both parties toward non-cooperation. When the increase reaches a certain limit, the evolutionary path of the system will develop qualitative changes and eventually tend toward non-cooperation.

(2)   The influence of operating cost coefficient on the evolutionary game model

When the initial cooperation willingness of both parties in the game is x = 0.45 y = 0.60 and other conditions remain unchanged, it can be seen from the evolutionary simulation results in Figure 12 that when the third-party logistics company M enjoys 55% of excess returns and bears 55% of operating costs, the two parties can reach a cooperative consensus on joint distribution. When the third-party logistics company M enjoys 55% excess income but reduces the share of cost borne, the third-party logistics company N is less willing to cooperate, and the cooperation will eventually be difficult to maintain, leading to a comprehensive breakdown.

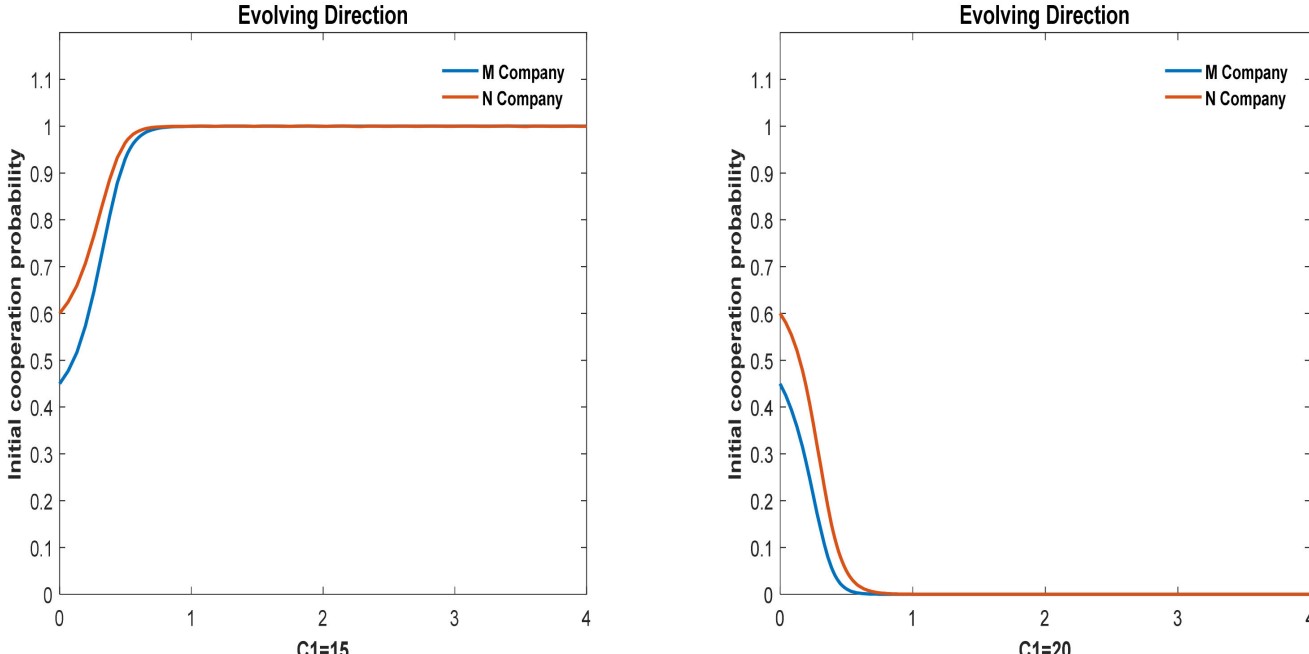

**Figure 11.** Managing operating costs for the influence of system evolution results.

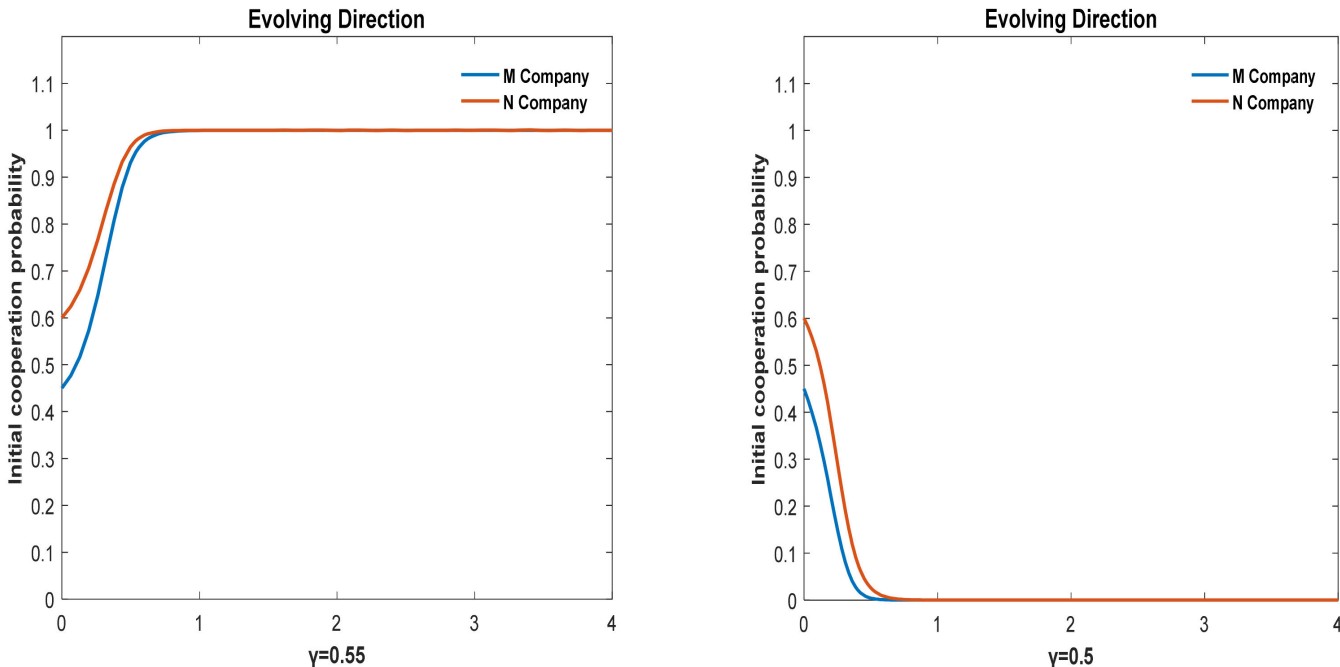

**Figure 12.** Influence of late operating cost coefficient on system evolution results.

### 4.2.5. Influence of Cooperation Risk-Bearing Coefficient

When the initial willingness to cooperate between the two parties of the game is $x = 0.45$ $y = 0.60$ and other conditions remain unchanged, the simulation of the evolution result shows the influence of the evolution result in Figure 13. It can be seen that when the cooperation risk bearing is 0.3, both parties can accept the cooperation risk within the cost-bearing range. However, when the cooperation risk increases to 0.5, the game parties will eventually choose not to cooperate in the long-term strategy repetition due to the high risk of cooperation failure, and the system will tend toward the comprehensive competition.

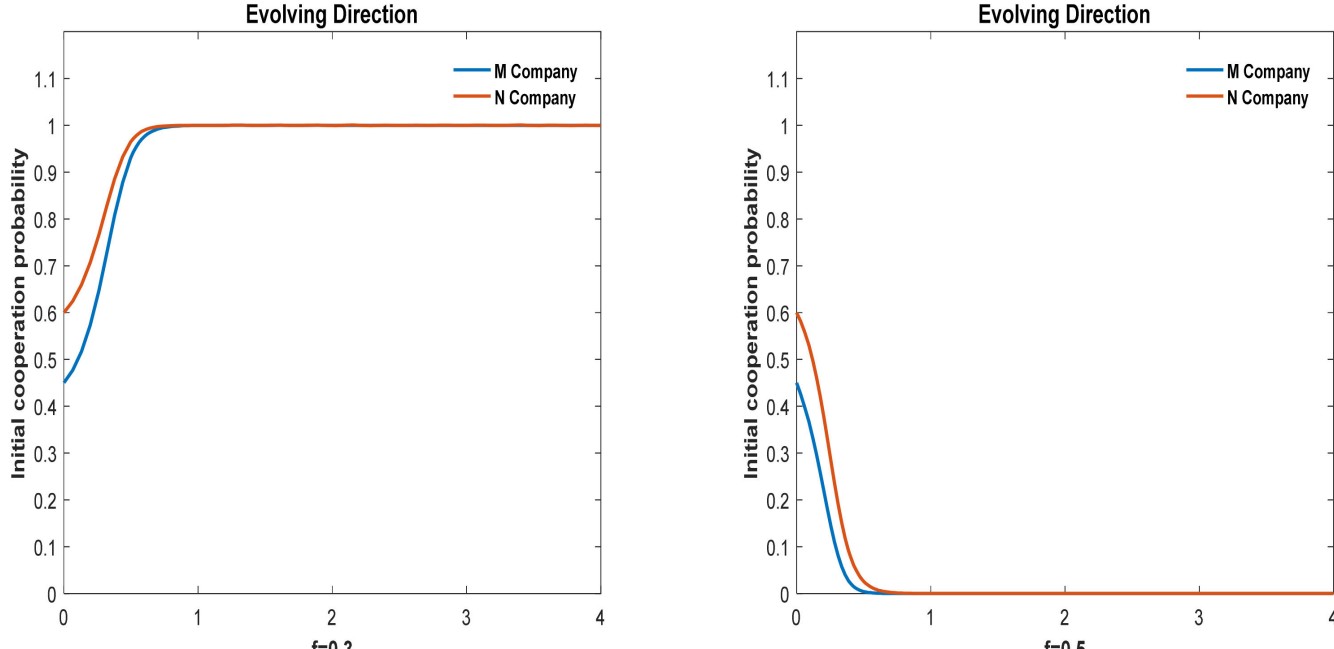

**Figure 13.** Influence of the cooperation risk-bearing coefficient on system evolution results.

### 4.2.6. Influence of Learning Absorption Capacity

The initial willingness to cooperate on both sides of the game is x = 0.45 y = 0.60 and the other conditions remain unchanged, it can be seen from the evolutionary results in Figure 14 that the learning and absorbing ability of both sides of the game is not very sensitive to the influence of the evolution results. When $L_M = 1.5\ L_N = 1.3$, the two sides tend to cooperate fully in the strategy. When $L_M = 1.2\ L_N = 1.6$, the learning and absorbing ability of one side decreases, the other side increases and the system eventually evolves into the full cooperation strategy, although it takes longer than before for both sides to converge to the full cooperation strategy. When $L_M = 1\ L_N = 2$, the difference in power between the two sides of the game is too large, and the evolutionary outcome will change qualitatively. In a realistic sense, mutually beneficial cooperation can be reached between enterprises with alternative business and between large-scale logistics enterprises and small-scale logistics enterprises. Large enterprises can take the lead in realizing the joint distribution of rural express, while maintaining the fairness of income distribution and cost allocation within the cooperation, which can accelerate the realization of the goal.

### 4.2.7. Influence of Positive External Benefits

When the initial willingness to cooperate between the two parties of the game is x = 0.45 y = 0.60 and other conditions remain the same. It can be seen from the evolutionary result Figure 15 that when $E_M = 20\ E_N = 18$, the system eventually tends toward the full cooperation strategy; when $E_M = 30\ E_N = 20$, the external economic benefits that both parties can obtain from each other by adopting their own competitive strategies will change from less to more, and the system will eventually tend toward the comprehensive competition and cooperation is difficult to achieve.

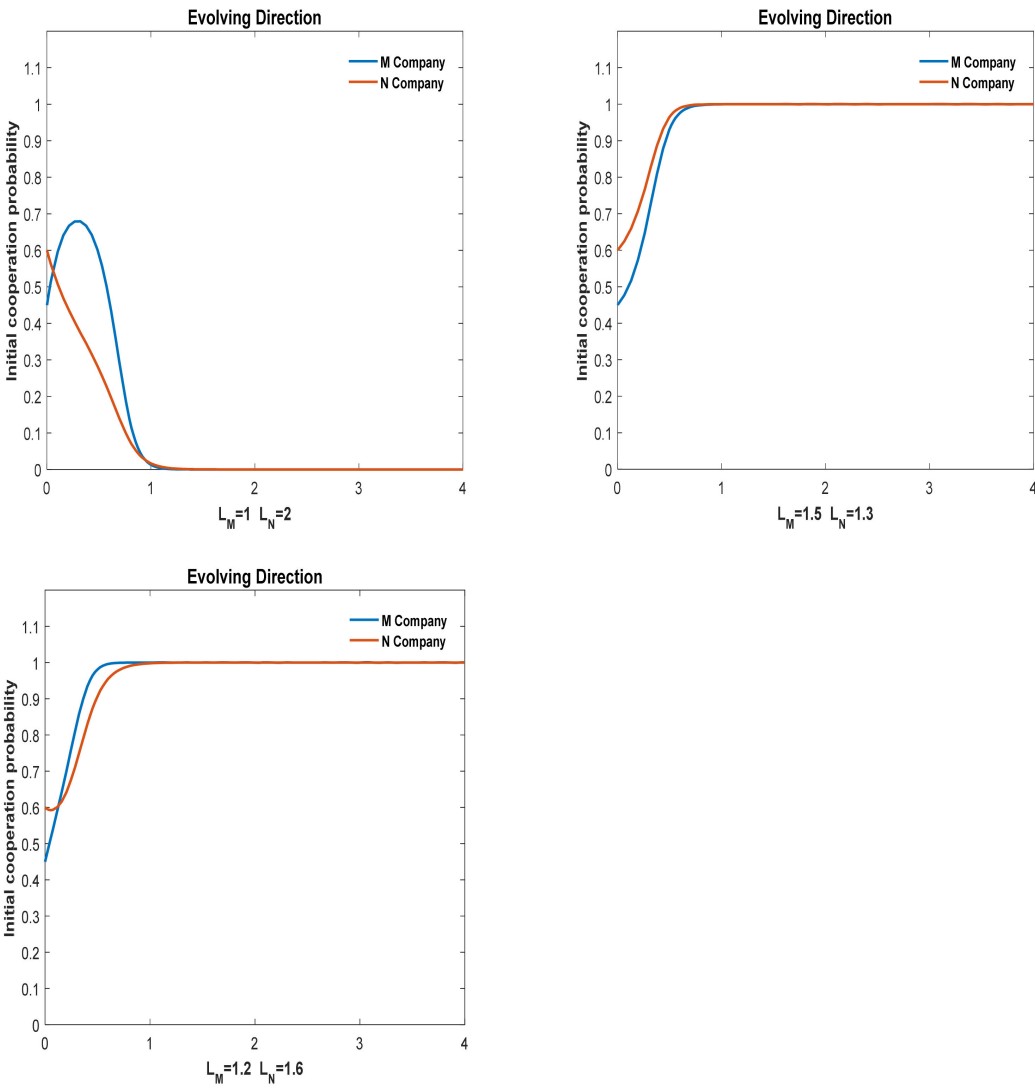

**Figure 14.** Influence of learning absorption capacity on system evolution results.

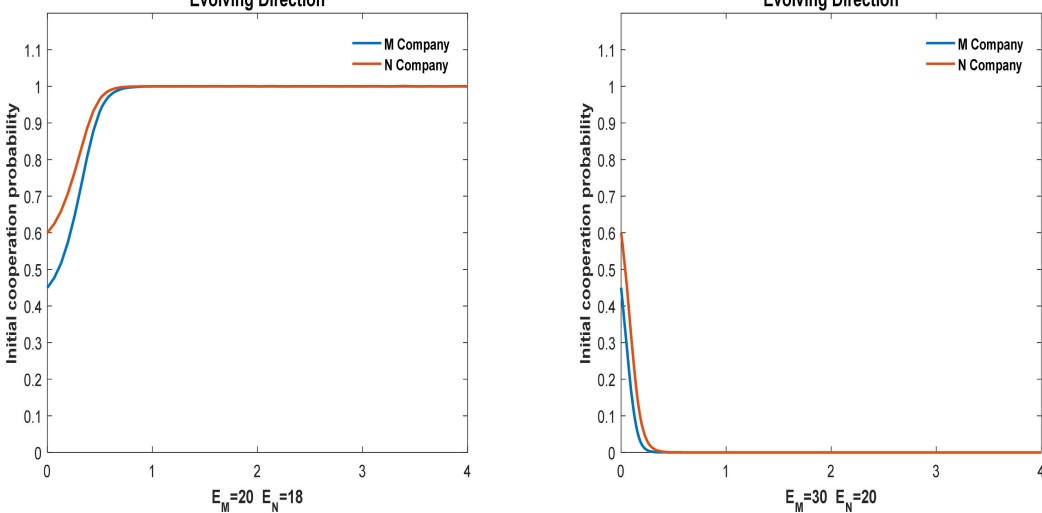

**Figure 15.** Influence of cooperation risk coefficient on system evolution results.

### 4.2.8. Impact of Default Penalty Cost

When the initial willingness to cooperate between the two parties of the game is x = 0.45 y = 0.60 and other conditions remain unchanged, it can be seen from the evolutionary result Figure 16 that when P = 15, the system eventually tends toward a full cooperation strategy. When the default penalty cost is reduced to P = 10, based on the psychology that the cost of cooperative speculation is reduced and the loss of default is reduced, the speculation willingness of both sides of the game will increase, and eventually the whole system will move toward comprehensive competition. It can be seen that the evolutionary path of the system is sensitive to the amount of protocol penalty.

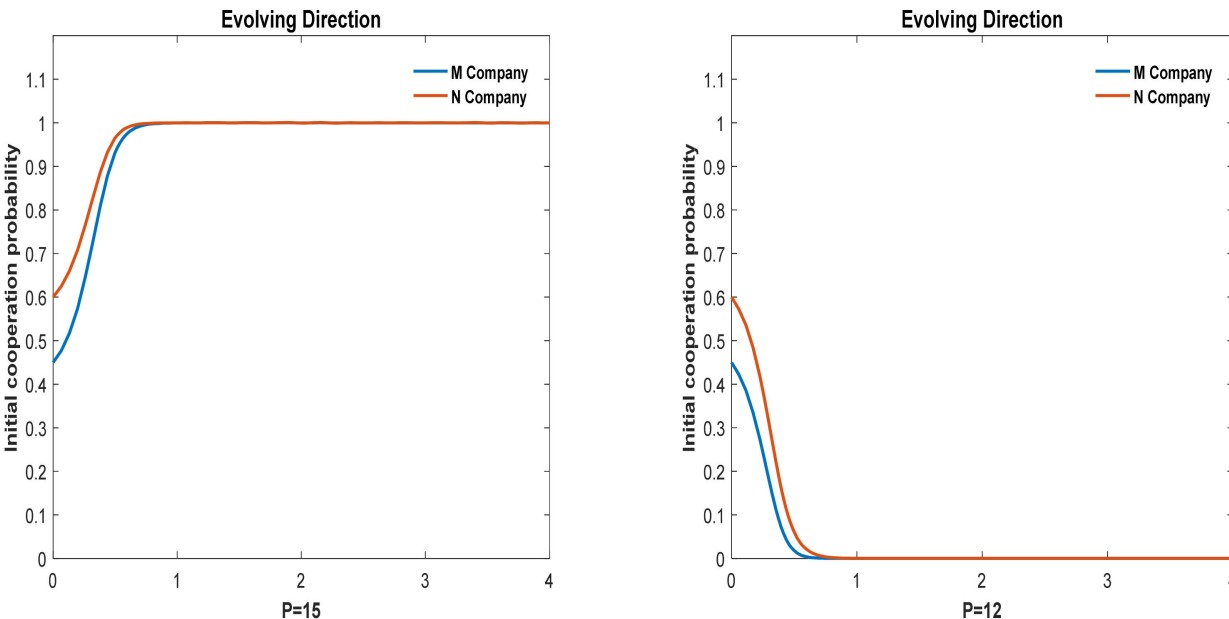

**Figure 16.** Influence of default penalty costs on system evolution results.

### 4.2.9. Influence of Incentive and Punishment Factors Introduced into e-Commerce Platforms

In Section 2.2, multiple parameters affecting rural express delivery enterprises were modeled and solved. Now, the reward and punishment factors of the e-commerce platform are introduced into the evolutionary game model and the values of the reward and punishment factors are assigned. There are six groups of related parameter values, as shown in Table 6.

**Table 6.** Value assignment table of reward and punishment factors on e-commerce platforms.

| Set No. | Group 1 | Group 2 | Group 3 | Group 4 | Group 5 | Group 6 |
| --- | --- | --- | --- | --- | --- | --- |
| $T_M$, $T_N$ | 0, 0 | 0, 0 | 0, 0 | 5, 5 | 10, 10 | 10, 5 |
| $F_M$, $F_N$ | 0, 0 | 5, 5 | 10, 10 | 0, 0 | 0, 0 | 10, 5 |

When the initial cooperation intention of both sides of the game is x = 0.45 y = 0.60 and other conditions remain the same, the evolutionary results in Figure 17 show that if no e-commerce platform rewards and penalties are introduced, the final result of the evolutionary game is that the express companies all choose competitive strategies. When only e-commerce platform rewards are introduced, the higher the reward, the faster the express company will choose cooperation strategies. When only e-commerce platforms are punished, the greater the government's punishment, the faster the express company will choose cooperation strategies. When the rewards and punishments of the e-commerce platform are introduced at the same time, the shorter the time for the express company to finally make a cooperative strategy. Therefore, the reasonable and effective reward and

punishment mechanism of the e-commerce platform will promote the joint distribution of express enterprises faster.

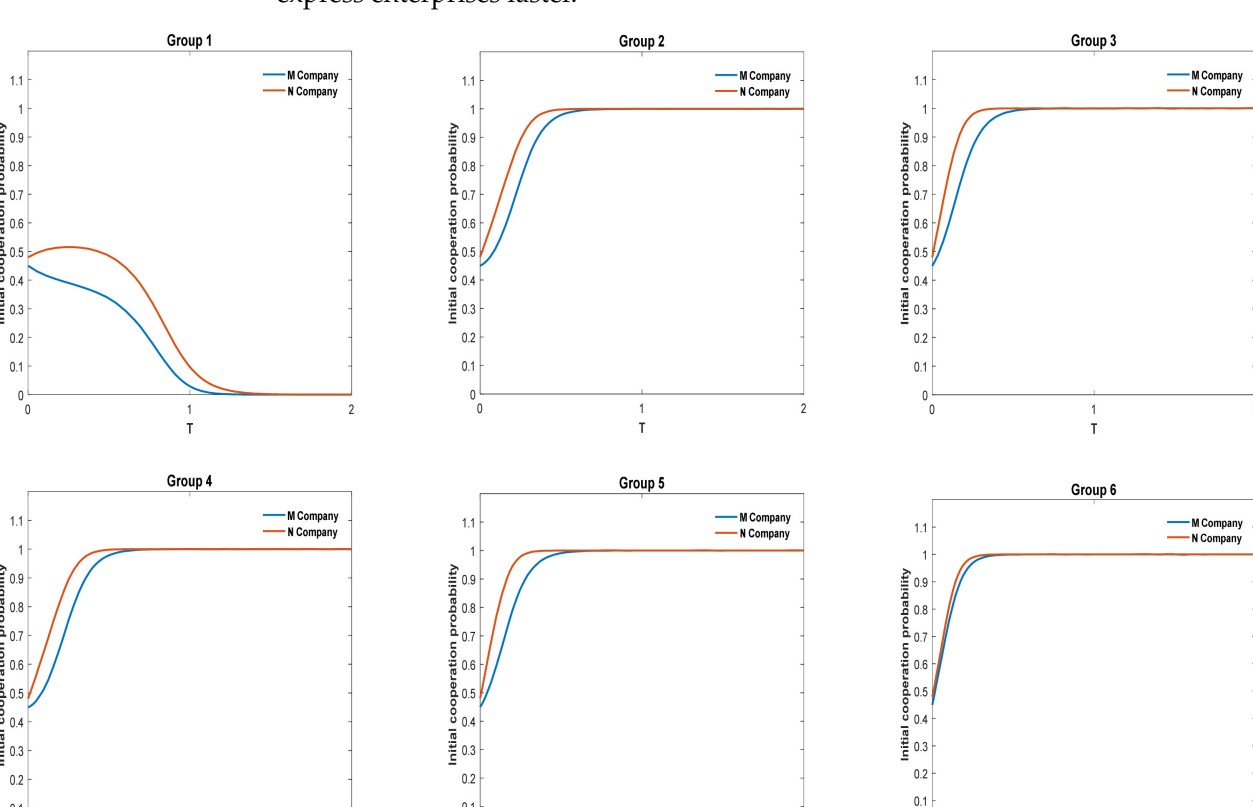

**Figure 17.** Influence of reward and punishment factors on the system evolution results of different platforms.

## 5. Conclusions

Based on the characteristics of express delivery in rural areas, this paper constructs the evolutionary game model and evolutionary simulation model of the joint distribution system of rural express delivery enterprises, analyzes the operation mechanism of the joint distribution system of rural express delivery enterprises and the evolution process of cooperation stability and draws the following conclusions.

(1) In the face of the current situation of rural express delivery, the establishment of a joint distribution alliance between express delivery enterprises in rural areas can introduce resource sharing and mutual benefit. In the joint distribution alliance, there are complicated competition and cooperation scenarios between different express companies, and the distribution of income and cost sharing become the key to the joint distribution system. Therefore, rural logistics enterprises will comprehensively weigh the benefits and costs of participating in the alliance, as well as their development strategies, so as to make alliance-cooperation decision-making behavior.

(2) There are five equilibrium points in the long-term evolution of the rural logistics joint distribution alliance system, and only two evolutionary stability strategies. When the Jacobian matrix meets $\det J > 0$ and $\text{tr} J < 0$, the equilibrium point of the joint distribution alliance system can reach the local stable state.

When $\Delta R > \text{Max}\left[C_M + P - \frac{1}{\alpha}L_M E_M, C_N + P - \frac{1}{1-\alpha}L_N E_N\right]$, the excess income obtained by a single enterprise within the alliance member is much greater than the sum of the enterprise's input cost, alliance penalty and effect loss, and express delivery enterprises have the source power to participate in the joint distribution alliance. If not, the alliance will

tend to break down. The operating conditions of logistics enterprises in the alliance will determine the saddle point value of the alliance and then directly affect the evolutionary path of the alliance.

(3) The stability of rural logistics joint distribution alliance is comprehensively affected by multiple factors such as alliance excess returns, alliance fines, positive effect loss, initial investment and operating costs; the influence mechanism is also different.

(4) As an important part of rural express joint distribution alliance, the e-commerce platform is of great significance for the implementation of joint distribution alliance formation. In the article, e-commerce platforms are not regarded as participants in building a three-way evolutionary game, but as an independent factor to influence the strategy and path selection of the evolutionary game. This method helps to explore the sensitivity of e-commerce platforms' rewards and punishments to evolutionary game models. The results show that e-commerce platforms will guide express companies to implement joint distribution through reasonable and effective reward and punishment measures to achieve the stable development of a joint distribution alliance.

The development of the rural logistics industry not only needs the participation of express delivery enterprises and e-commerce platforms, but also needs the participation of the government and consumers. Therefore, in order to achieve the stable development of a joint distribution alliance, we need two aspects. For example, the internal factors of express enterprises should coordinate and handle the cost allocation and revenue sharing mechanism, constantly improve the business operation capacity of the alliance, give play to the alliance's synergy efficiency, obtain excess profits and enhance the overall competitiveness and sustainable development ability of the alliance. In contrast, due to social and environmental factors, the government promulgates relevant systems and policies to support the logistics industry, master the direction and do a good job in macro-control. Consumers actively participate in the information feedback, which makes the whole distribution process form a benign closed-loop.

**Author Contributions:** Methodology, H.Z.; Investigation, H.Z.; Resources, H.Z.; Data curation, H.Z.; Writing – original draft, H.Z.; Supervision, M.L.; Project administration, M.L.; Funding acquisition, M.L. All authors have read and agreed to the published version of the manuscript.

**Funding:** This research was funded by the Natural Science Foundation of China (grant No. 71971130), Research on the Construction Path and Effect Evaluation System of First-Class Undergraduate Majors in Shandong Province (grant No. 2021JXY058) and Research on the Development Trend and Countermeasures of Industrial Internet in Shandong Province (grant No. 2021RZB02007).

**Institutional Review Board Statement:** Not applicable.

**Informed Consent Statement:** Not applicable.

**Data Availability Statement:** The data presented in this study are available in [Strategy selection and Research of evolutionary game problems].

**Conflicts of Interest:** The authors declare no conflict of interest.

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
