# Peer review of "Study on Joint Distribution Mode and Evolutionary Game of Express Enterprises in Rural Areas"

_sustainability, doi:10.3390/su15021520_

Round 1

Reviewer 1 Report

The content of this paper is rich, the numerical analysis is logical, and the conclusion is convincing. But there are still some typos and inconsistent statements that need to be optimized. Therefore, I recommend that this paper be accepted after minor revision.

Author Response

Dear Editor:
I am writing to submit our manuscript entitled, " Study on Joint Distribution Mode and Evolutionary Game of Express Enterprises in Rural Areas ", which we wish to be considered for publication in “Sustainability”.Thank you for your suggestions on the content of my paper, I have revised and supplemented each of these issues. Please check the attachment for details.

Thank you and best regards.

Reviewer 2 Report

Although there are statements in the introduction that the improvements in the rural express distribution system will positively affect the rural economy, the contribution of the joint distribution model to rural development has not been sufficiently addressed. It would be useful to emphasize the importance of the study by mentioning this issue further. In Part 2, which describes the model structure and solution, has almost no references. For example, definition sentences describing the system evolutionary game model and stability analysis. Also, findings are not discussed in the literature at all. As stated in the introduction, although it is difficult to carry out an in-depth discussion since there are few studies in this field, the findings obtained in this study can be discussed.

Author Response

(The authors gave the same response as above.)

Reviewer 3 Report

High delivery cost and low efficiency of courier delivery are outstanding problems in rural areas of China. This manuscript, taking courier delivery enterprises in rural areas as the research object, constructed a three-level “county-town-village” joint distribution system in which e-commerce platforms participate and established an evolutionary game model of courier delivery enterprise joint distribution alliance to analyze the model using numerical simulation, which has important value to the healthy development of courier delivery in rural China, and I would comment it be accepted after minor revision.

spcific suggestions are as follows:

1. The core viewpoints of the achievements on co-delivery mode and evolutionary game at home and abroad need to be refined focusing on the content, method or scale rather than the listing of the literatures.

2. The construction of a Three-Level Joint Delivery System of Rural Express and numerical simulation is nainly based on evolutionary game of two express companies. What's going on of the evolutionary game if there are multiple delivery companies? it should be discussed appropriately in the results analysis.

3. In fact, the model of rural co-delivery and evolutionary game constructed in this manucript are theoretical models. Is there any typical case that can support the practice of rural co-delivery development in China?

Author Response

(The authors gave the same response as above.)
